# Earliest modern human genomes constrain timing of Neanderthal admixture

Arev P. Sümer[1✉], Hélène Rougier[2], Vanessa Villalba-Mouco[1,3], Yilei Huang[1,4], Leonardo N. M. Iasi[1], Elena Essel[1], Alba Bossoms Mesa[1], Anja Furtwaengler[1], Stéphane Peyrégne[1], Cesare de Filippo[1], Adam B. Rohrlach[1,5], Federica Pierini[1], Fabrizio Mafessoni[6], Helen Fewlass[1,7,8], Elena I. Zavala[1,9], Dorothea Mylopotamitaki[1,10], Raffaela A. Bianco[1], Anna Schmidt[1], Julia Zorn[1], Birgit Nickel[1], Anna Patova[1], Cosimo Posth[11], Geoff M. Smith[1,12], Karen Ruebens[1,10,12], Virginie Sinet-Mathiot[1,13,14], Alexander Stoessel[1,15], Holger Dietl[16], Jörg Orschiedt[16,17], Janet Kelso[1], Hugo Zeberg[1,18], Kirsten I. Bos[1], Frido Welker[19], Marcel Weiss[1,20], Shannon P. McPherron[1], Tim Schüler[21], Jean-Jacques Hublin[1,10], Petr Velemínský[22], Jaroslav Brůžek[23], Benjamin M. Peter[1,24], Matthias Meyer[1], Harald Meller[16], Harald Ringbauer[1], Mateja Hajdinjak[1], Kay Prüfer[1,25✉] & Johannes Krause[1,25✉]

Modern humans arrived in Europe more than 45,000 years ago, overlapping at least 5,000 years with Neanderthals[1–4]. Limited genomic data from these early modern humans have shown that at least two genetically distinct groups inhabited Europe, represented by Zlatý kůň, Czechia[3] and Bacho Kiro, Bulgaria[2]. Here we deepen our understanding of early modern humans by analysing one high-coverage genome and five low-coverage genomes from approximately 45,000-year-old remains from Ilsenhöhle in Ranis, Germany[4], and a further high-coverage genome from Zlatý kůň. We show that distant familial relationships link the Ranis and Zlatý kůň individuals and that they were part of the same small, isolated population that represents the deepest known split from the Out-of-Africa lineage. Ranis genomes harbour Neanderthal segments that originate from a single admixture event shared with all non-Africans that we date to approximately 45,000–49,000 years ago. This implies that ancestors of all non-Africans sequenced so far resided in a common population at this time, and further suggests that modern human remains older than 50,000 years from outside Africa represent different non-African populations.

Neanderthals lived in Europe and western Asia for hundreds of thousands of years before their disappearance around 40 thousand years ago (ka)[1,5]. In the last few thousand years of their documented existence, Neanderthals met and interbred with modern humans who arrived from Africa, and, as a result, 2–3% of the ancestry of present-day non-Africans derives from Neanderthals[6].

So far, only five sites have yielded genome-wide data from modern humans who lived before 40 ka and thus temporally overlapped with Neanderthals (Fig. 1). The Neanderthal ancestry in the genomes from two of these sites probably originated from just a single introgression event (that is, an admixture with Neanderthals that may have continued over several generations)[3,7]. However, the genomes of individuals from the other three sites showed evidence for additional, more recent Neanderthal introgression events[2,8,9]. The high-coverage genome of the approximately 44-thousand year (kyr)-old Ust'-Ishim individual,

an early inhabitant of Siberia, shows signals for such an additional introgression event around 30–50 generations before the individual lived[3,8,9]. A similar analysis has shown that the approximately 40-kyr-old Oase 1 individual from Peștera cu Oase, Romania, and four individuals dating to around 44 ka from Bacho Kiro, Bulgaria[2], had Neanderthal ancestors probably within the last 10–20 generations before they lived. By contrast, no evidence for additional admixtures has been found for the 40-kyr-old Tianyuan individual from China[10] or the Zlatý kůň individual from Czechia[3]. Although direct radiocarbon dating yielded unreliable results for Zlatý kůň, the lengths of Neanderthal ancestry segments in the genome indicated an age of at least 45 kyr (ref. 3).

With the exception of the Tianyuan individual, who was part of the ancestral population of East Asians, all previously mentioned individuals showed no, or at most a limited, direct contribution to the ancestry of later Out-of-Africa populations. Notably, the Zlatý kůň individual

[1]Max Planck Institute for Evolutionary Anthropology, Leipzig, Germany. [2]California State University Northridge, Northridge, CA, USA. [3]Institute of Evolutionary Biology, CSIC-Universitat Pompeu Fabra, Barcelona, Spain. [4]Institute of Computer Science, Universität Leipzig, Leipzig, Germany. [5]School of Biological Sciences, University of Adelaide, Adelaide, South Australia, Australia. [6]Weizmann Institute of Science, Tel Aviv, Israel. [7]Francis Crick Institute, London, UK. [8]University of Bristol, Bristol, UK. [9]University of California, Berkeley, CA, USA. [10]Chaire de Paléoanthropologie, CIRB, Collège de France, Paris, France. [11]Tübingen University, Tübingen, Germany. [12]Department of Archaeology, University of Reading, Reading, UK. [13]University of Bordeaux, CNRS, Ministère de la Culture, PACEA, Pessac, France. [14]University of Bordeaux, CNRS, Bordeaux INP, CBMN, UMR 5248 and Bordeaux Proteome Platform, Bordeaux, France. [15]Friedrich Schiller University Jena, Institute of Zoology and Evolutionary Research, Jena, Germany. [16]Landesamt für Denkmalpflege und Archäologie Sachsen-Anhalt-Landesmuseum für Vorgeschichte, Halle, Germany. [17]Prähistorische Archäologie, Freie Universität, Berlin, Germany. [18]Karolinska Institutet, Stockholm, Sweden. [19]Globe Institute, University of Copenhagen, Copenhagen, Denmark. [20]Friedrich-Alexander-Universität Erlangen-Nürnberg, Institut für Ur- und Frühgeschichte, Erlangen, Germany. [21]Thuringian State Office for the Preservation of Historical Monuments and Archaeology, Weimar, Germany. [22]National Museum, Prague, Czechia. [23]Charles University, Prague, Czechia. [24]University of Rochester, Rochester, NY, USA. [25]These authors contributed equally: Kay Prüfer, Johannes Krause. ✉e-mail: arev_suemer@eva.mpg.de; pruefer@eva.mpg.de; krause@eva.mpg.de

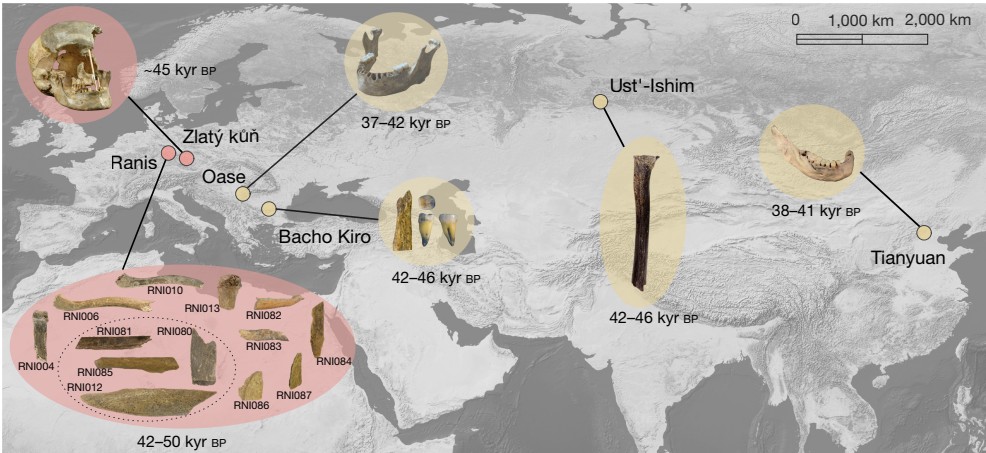

**Fig. 1 | Geographical distribution of modern human specimens older than 40 kyr that produced genome-wide data.** Specimens with new genome-wide data produced in this study are shaded in pink. Ages give 95.4% CIs on calibrated radiocarbon dates except for Zlatý kůň, for which we give the estimated age on the basis of Neanderthal segment lengths. The Ranis specimens within the dashed circle belong to the same individual. Credits: photographs of the Ranis specimens are adapted from ref. 4, Springer Nature Limited, under a Creative Commons licence, CC BY 4.0; the photograph of the Zlatý kůň skull is adapted from the Department of Anthropology, National History Museum of Prague (photographer: Marek Jantač); photographs of the Oase and Bacho Kiro specimens, © MPI-EVA/Rosen Spasov (from www.mpg.de/16663512/genomes-earliest-europeans), and of a 40-kyr-old modern human jawbone, © MPI for Evolutionary Anthropology/Svante Pääbo (from www.mpg.de/9278783/modern-humans-neandertals-interbreeding-europe); photograph of the Ust'-Ishim specimen is adapted from ref. 9, Springer Nature Limited, under a Creative Commons licence, CC BY 4.0; the photograph of the Tianyuan specimen is reproduced from www.science.org/content/article/last-ice-age-wiped-out-people-east-asia-well-europe (Shaoguang Zhang/Institute of Vertebrate Paleontology and Paleoanthropology). The base map was made with Natural Earth (www.naturalearthdata.com/).

belonged to a deeply divergent population that separated from the lineage leading to non-Africans earlier than any other known ancient or present-day Out-of-Africa population and is at present the only representative of this early branch[3].

Although distinct stone tool technologies existed 40–50 ka in Europe[11,12], only one of the five early modern human sites that yielded genome-wide data is associated with such a technology—namely, the Initial Upper Palaeolithic (Bachokirian) in Bacho Kiro[2]. It is therefore still heavily debated which stone tool technocomplexes were made by early modern humans and which by Neanderthal groups[12]. Recently, the Lincombian–Ranisian–Jerzmanowician (LRJ) technocomplex, which was present in central and northwestern Europe around 41–47.5 ka (refs. 4,13), has been shown to be made by early modern humans[4]. This assignment was based on the study of mitochondrial DNA (mtDNA) from 11 bone fragments that were found at Ilsenhöhle in Ranis, Germany (hereafter, 'Ranis'), in association with the LRJ during two separate excavations of the site (1932–1938 and 2016–2022). These bone fragments were directly radiocarbon-dated to between 42,200 and 49,540 calibrated years before the present (cal BP) (95.4% probability; Supplementary Information 2).

To determine the relationship of the Ranis individuals to other human populations and to gain further insights into the genetic history of those early modern humans, we first assessed DNA preservation by generating shallow shotgun sequence data from 13 specimens from Ranis. Data from two of these specimens were excluded from downstream analyses because they showed a high fraction of contaminating DNA sequences from present-day humans (greater than 40% contamination in a test based on characteristic patterns of ancient DNA damage[14]; Supplementary Table 1.2 and Supplementary Information 3). However, one specimen showed exceptional preservation (30% of DNA sequences originated from the ancient individual; Supplementary Table 1.2). We therefore produced further shotgun sequence data from this specimen resulting in a 24-fold coverage genome. We also generated further sequence data for Zlatý kůň on the basis of newly constructed and existing DNA libraries, resulting in a 20-fold coverage genome. We estimated approximately 3% present-day human contamination in the data of both high-coverage genomes (Supplementary Table 4.1).

Ranis specimens were subjected to targeted enrichment for 1.2 million variants suitable for population genetic analysis (1240k array)[15] and 1.7 million variants informative for Neanderthal and Denisovan ancestry (Archaic Admixture array)[16]. Two of these specimens were excluded because they yielded insufficient data (less than 0.02×) or were highly contaminated (greater than 20%) (Supplementary Tables 3.1–3.3). We further restricted our analysis to sequences carrying characteristic patterns of ancient DNA damage to reduce contamination for five of nine specimens with an estimate of over 5% (Supplementary Table 3.2). Using a method that estimates contamination using regions with no differences between the parental copies of the genome[17], we estimate that for all nine specimens the filtered data contain less than 5% human contamination (Supplementary Table 3.2).

## Kinship and uniparental markers

Multiple Ranis specimens share identical mtDNA and could potentially represent the same individual[4]. To determine how many different individuals are represented by the nine Ranis specimens, we used autosomal and X-chromosome data to infer the genetic sex and biological kinship of all individuals (Supplementary Information 6). We find that five specimens originate from separate individuals (Ranis4, Ranis6, Ranis10, Ranis87 and the high-coverage-genome-yielding Ranis13), whereas the remaining four specimens all belonged to the same individual (Ranis12) (Fig. 2a). Three of the four Ranis12 specimens were discovered during the original 1932–1938 excavation, whereas one specimen was found in the recent 2016–2022 excavation. These bone fragments were distributed over a large area and link the stratigraphic sequences from the two excavations (Extended Data Fig. 1).

As expected for separate samples from the same individual, all four specimens of Ranis12 had the same genetic sex (female) and carried identical mtDNA. Another individual, Ranis10, also shares the same mtDNA sequence; however, this individual is a male and does not seem to be a close kin to Ranis12 (Fig. 2a). Ranis4 and Ranis6 were also identified in the previous study[4] to have identical mtDNA. Our analysis indicates that both of these individuals are female and show a parent–offspring relationship. Ranis6 is a juvenile younger than 5 years on the

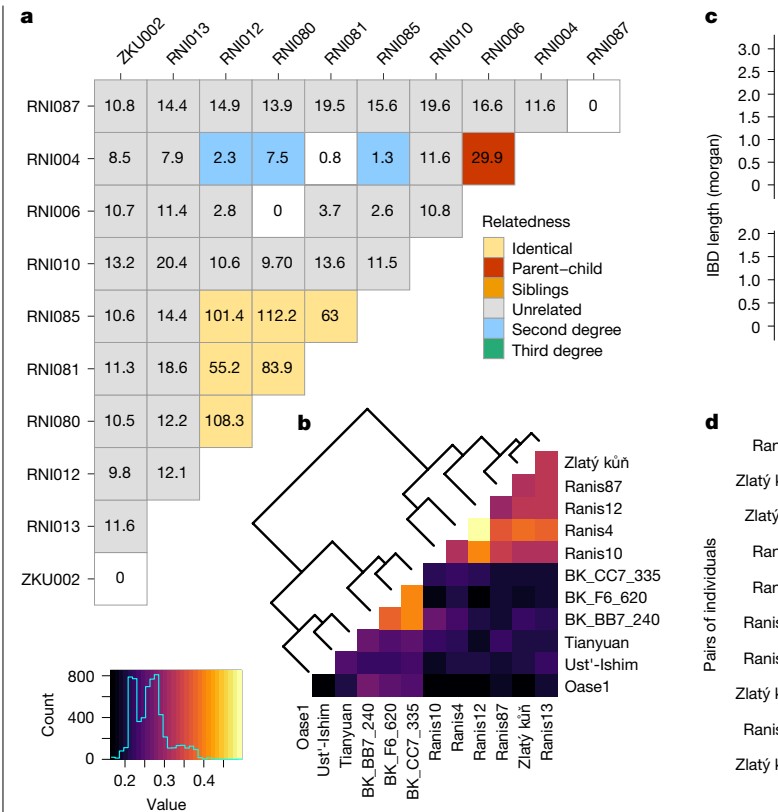

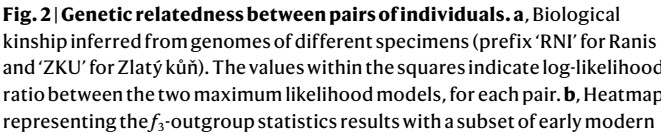

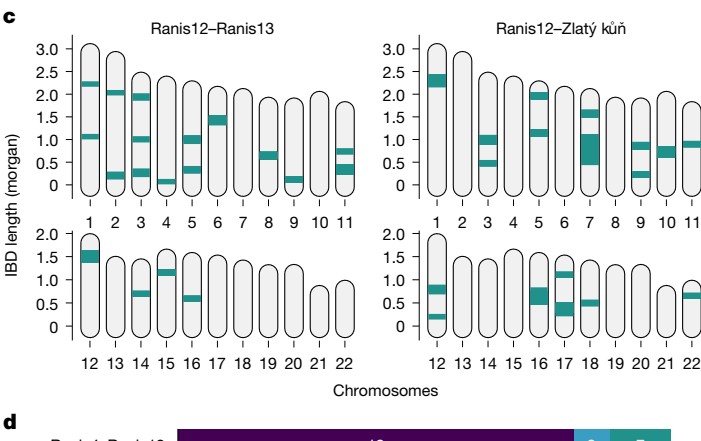

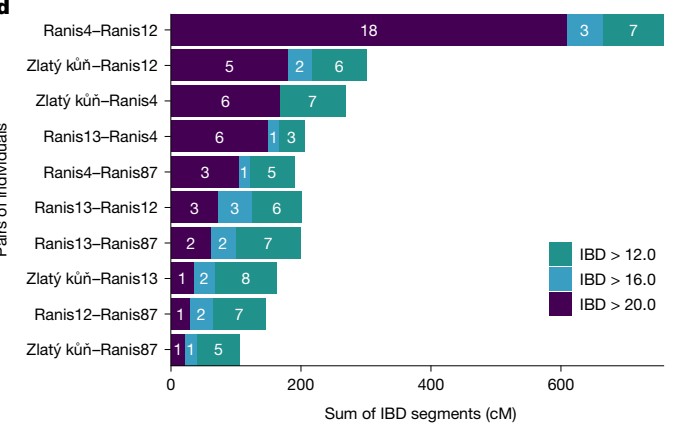

**Fig. 2 | Genetic relatedness between pairs of individuals. a**, Biological kinship inferred from genomes of different specimens (prefix 'RNI' for Ranis and 'ZKU' for Zlatý kůň). The values within the squares indicate log-likelihood ratio between the two maximum likelihood models, for each pair. **b**, Heatmap representing the $f_3$-outgroup statistics results with a subset of early modern humans older than 40 kyr. **c**, Pairwise IBD sharing (greater than 12 cM) for comparisons between individuals Ranis12–Ranis13 and Ranis12–Zlatý kůň. **d**, Total IBD sharing in three different length categories plotted for the ten Ranis/Zlatý kůň pairs with the highest IBD sharing. Ranis6 and Ranis10 were excluded due to low coverage.

basis of the size of the specimen, implying that Ranis4 and Ranis6 were mother and daughter. A more distant second- or third-degree relationship was also detected between Ranis4 and Ranis12 (Supplementary Information 6).

Three of the six individuals (Ranis13, Ranis87 and Ranis10) were identified as male. We assigned the Y-chromosomal haplogroup from capture and shotgun sequencing data to a basal haplogroup F for Ranis10, whereas Ranis13 and Ranis87 were assigned to haplogroup NO (K2a), similar to the Siberian ancient individual Ust'-Ishim (Supplementary Fig. 18.1). No evidence of biological kinship was detected among the three male individuals.

## Linking Zlatý kůň and Ranis

Taken together, these results show that the Ranis individuals belonged, at least partially, to a group closely linked by familial relationships. However, the kinship analysis applied here is able to reliably detect relationships only up to the third degree[18]. To infer more distant relatedness, we used a method to detect segments of shared genetic ancestry (identical by descent (IBD))[19] among the genomes of four Ranis individuals with sufficient coverage and Zlatý kůň (Supplementary Information 8). Congruent with the previously detected kinship of Ranis4 and Ranis12, the IBD analysis indicates a relationship of third to fourth degree on the basis of shared ancestry segments (Extended Data Fig. 2). However, surprisingly, the sum of long (greater than 12 centimorgans (cM)) shared segments between the Zlatý kůň individual and Ranis12 (301 cM total), and between Zlatý kůň and Ranis4 (268 cM total), exceeded the sum of such long segments shared among other Ranis individuals (less than 205 cM total) (Fig. 2c and Supplementary Table 8.1). The length

and frequency of these segments indicate that Zlatý kůň was a fifth- or sixth-degree relative of Ranis12 and Ranis4, and more distantly related to the other Ranis individuals (Extended Data Fig. 2).

Zlatý kůň and some Ranis individuals shared ancestors in their recent family history. It is therefore conceivable that all of these individuals originate from the same population. We tested this hypothesis by calculating pairwise $f_3$-outgroup statistics[20] in comparison with other pre-40-kyr-old hunter-gatherers (Extended Data Fig. 3). Pairwise comparisons between Zlatý kůň and Ranis individuals yielded consistently higher values than comparisons of these individuals with other hunter-gatherers, indicating that they were part of the same population. We merged the data of all Ranis individuals and computed $f_4$-statistics[20] to test whether the combined data of Ranis individuals or Zlatý kůň share more alleles with ancient hunter-gatherers. As expected from a model in which Zlatý kůň and Ranis form a uniform group, we found no significant differences in sharing (Extended Data Fig. 4a,b and Supplementary Figs. 9.3 and 9.4). We also tested the population relationship using only the Ranis13 and Zlatý kůň high-coverage genomes with qpGraph[20]. Models that would place the individuals with one ancestral to the other provided a poorer fit to the data than a model establishing a common branch for both. This common branch of Ranis13 and Zlatý kůň separates earlier from the ancestral Out-of-Africa lineage than the lineage leading to the Bacho Kiro individuals (Extended Data Fig. 5). The population corresponding to this common branch also splits earlier from the Out-of-Africa lineage than the population represented by the Ust'-Ishim genome in a combined model using the inference method momi2 (ref. 21) (Extended Data Fig. 4c). Ranis and Zlatý kůň are thus members of the same population, which we refer to as the Zlatý kůň/ Ranis population hereafter.

## Population continuity

To test whether later populations derive ancestry directly from Zlatý kůň/Ranis, we measured allele-sharing using sites targeted by the 1240k capture array. In pairwise comparisons of hunter-gatherer groups, we found some evidence for significant sharing ($|Z| > 3$ for 151 of 2,701 comparisons of the form $f_4$(hunter-gatherer, hunter-gatherer; Ranis13, Mbuti)). However, it is known that some populations carry deeply divergent ancestry in their genomes that derives either from Africans or from an ancestral lineage that diverged from non-Africans before they received Neanderthal ancestry ('Basal Eurasian lineage'). In our comparisons we find that all significant results can be explained by these ancestries. Our results indicate that the Zlatý kůň/Ranis population did not contribute ancestry to later hunter-gatherers, confirming previous analyses using the low-coverage Zlatý kůň data[6] (Supplementary Fig. 9.5 and Supplementary Table 4.3). However, in contrast to these results, a recent study of two low-coverage genomes from 36–37-kyr-old Buran Kaya III individuals from Crimea found signals that were interpreted as evidence for a contribution of Zlatý kůň ancestry to these individuals[22]. In an attempt to maximize statistical power in the test, we included all sites covered by the combined data from Buran Kaya III in a comparison with the high-coverage Zlatý kůň, Ranis13 and Ust'-Ishim genomes, and low-coverage early European hunter-gatherer genomes from Russia[23,24]. We were unable to reproduce the signal of increased sharing with this dataset ($|Z| < 3$ in 75 comparisons of the form $D$(Kostenki14/Sughir1-4, Buran Kaya III, Zlatý kůň/Ranis13/Ust'-Ishim, African); Supplementary Fig. 9.14). Our results indicate that the Zlatý kůň/Ranis population shows no contribution to later Out-of-Africans, similar to Ust'-Ishim and Oase 1 (ref. 2).

## Population size

The early appearance of members of the Zlatý kůň/Ranis population in Europe, the absence of genetic continuity with later Europeans, as well as the close kinship between individuals even across archaeological sites suggest that this early modern human population was rather small. Several measures can be informative about the effective population sizes, corresponding to the number of breeding individuals per generation, at different times in an individual's past[25,26]. A broad estimate of the population size over longer periods of time can be inferred from the average number of sequence differences observed between the two sets of chromosomes that an individual inherits from their parents (heterozygosity). In contrast to Neanderthal genomes in which we observe approximately 2 heterozygous positions per 10,000 sites, the genomes of Ranis13 and Zlatý kůň show four times higher values of heterozygosity (approximately 7–8 of 10,000 sites) which are in line with estimates for other ancient and present-day modern humans outside Africa (Extended Data Fig. 6c and Supplementary Table 3.1). This long-term measure of population size contrasts with the short-term estimates based on regions of homozygosity (ROH), that is, regions with few to no heterozygous sites that are formed by consanguinity in the family history of an individual. Segments of ROH larger than 4 cM are observed for both Ranis13 and Zlatý kůň and sum to 5.2% and 7.2% of the genome, respectively (Extended Data Fig. 6a). A comparable value is found for the 45-kyr-old Vindija Neanderthal (7.0%), but both values are substantially higher than that observed for Ust'-Ishim (3.7%). Much of the consanguinity in the history of Ranis13 and Zlatý kůň seems to have occurred a few generations before these individuals lived, as 40–50% of the total ROH resides in segments of 12 cM or longer. However, this level of ROH is not due to a close family relationship between the parents of these individuals as has been observed for the Altai Neanderthal[27], but rather due to a small population size in their very recent past.

Long ROH can also be inferred from low-coverage data by analysing 1240k capture sites using the hapROH method[28]. In our analysis of the low-coverage Ranis genomes, we find broadly similar patterns of consanguinity which can be explained by a small population with a size of around 300 individuals (95% confidence interval (95% CI), 242–374) in addition to some level of recent inbreeding. This estimate largely reflects the population size within the last 50 generations (Supplementary Fig. 7.6 and Supplementary Table 8.1).

A small effective population size may negatively affect an individual's ability to adapt to pathogens. We therefore analysed the genetic diversity at the human leukocyte antigen (HLA) loci, which are among the most polymorphic loci in the human genome[29,30]. One of five HLA loci were homozygous in the Zlatý kůň genome and three of five in the Ranis13 genome. These loci do not fall into long ROH segments in the genome, which suggests that the homozygosity is probably caused by the small population size rather than consanguinity (Supplementary Tables 3.2–3 and 5.1). Similar levels of homozygosity in the HLA region are observed only for isolated present-day human populations (Supplementary Information 19).

Taken at face value, the level of heterozygosity seems to be at odds with the signals of consanguinity in the Ranis13 and Zlatý kůň genomes. However, the analysis of effective population size over time using the pairwise sequentially Markovian coalescent (PSMC) method[25] shows that both individuals lived shortly after a strong reduction in population size that has been associated with the Out-of-Africa event[31] (Supplementary Figs. 11.2 and 11.3). This reduction would explain the observed levels of heterozygosity which match those of other non-Africans. However, from our ROH analysis we infer that the Zlatý kůň/Ranis population experienced a further reduction in size when their ancestors migrated into Europe, which did not last long enough to substantially reduce overall heterozygosity. A small size for the Zlatý kůň/Ranis population is also supported by the sharing of IBD segments between the two high-coverage genomes, yielding an estimate of the effective population size within the last 15 generations of approximately 160 individuals (100–240 with 95% probability) (Extended Data Fig. 6b).

## Dating Zlatý kůň

Radiocarbon dating gives an age of around 45 ka for Ranis13 (ref. 4) whereas a reliable direct date for Zlatý kůň could not be established, probably because the Zlatý kůň skull was treated with adhesives derived from animal products[3]. However, on the basis of shared IBD segments, Ranis13 and Zlatý kůň probably lived three generations apart and confidently within at most 15 generations from each other, suggesting an age of around 45 ka for Zlatý kůň as well (Extended Data Fig. 6b). The high-coverage genomes of both individuals also allow us to estimate their age based solely on molecular data using two approaches: (1) an age estimate based on the lack of mutations in an ancient compared with a present-day genome[9,27] (Extended Data Fig. 7a), and (2) an estimate based on the shift in demographic histories estimated by PSMC from genomes of different ages[32] (Extended Data Fig. 7b–d). Applying these two approaches to the two high-coverage genomes resulted in point estimates of around 47–48 ka, albeit with large uncertainty (Supplementary Tables 10.1 and 11.4). In line with this age, we find that typical high-frequency phenotypic variants in present-day Europeans such as lactose tolerance, light pigmentation and lighter hair are absent from both the Ranis13 and Zlatý kůň genomes (Supplementary Information 20).

## Neanderthal ancestry and dating

All non-Africans trace a small fraction of their ancestry back to Neanderthals. To investigate Neanderthal ancestry in the Ranis and Zlatý kůň individuals, we used a segment-detection method (admixfrog[33] v.0.7.1) that inferred 2.9% Neanderthal ancestry (union of 95% CIs, 2.8–3.0%) for the high-coverage Ranis13 and Zlatý kůň genomes (Fig. 3a,b). The detected segments are on average

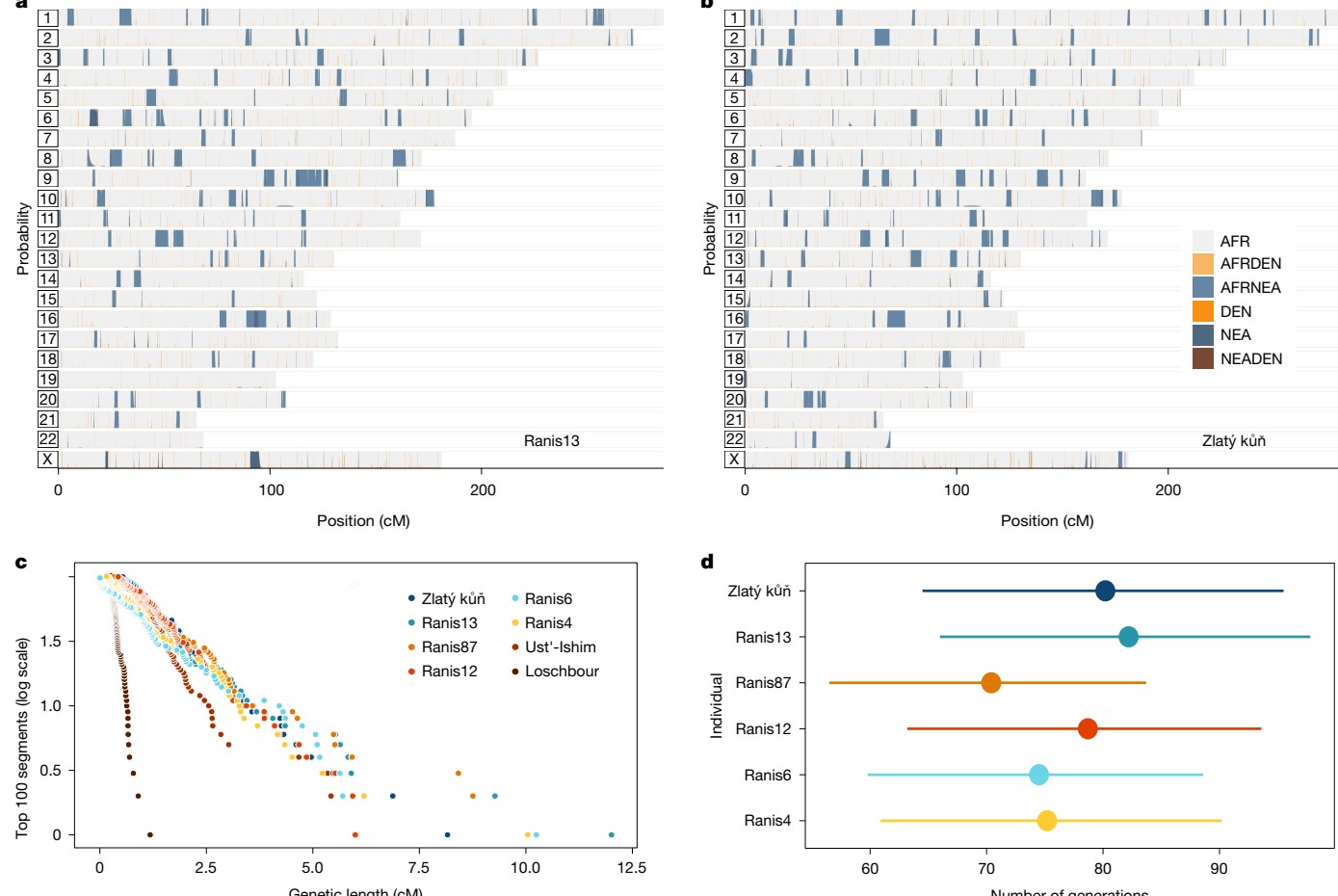

**Fig. 3 | Neanderthal ancestry. a,b**, Segments in Ranis13 (**a**) and segments in Zlatý kůň (**b**). Colours indicate the state of the called segment: grey for the homozygous African state (AFR), and dark orange and dark blue for homozygous Denisovan (DEN) and Neanderthal (NEA) states, respectively. The remaining three colours indicate heterozygous states as stated in the colour legend (AFRDEN for African/Denisovan, AFRNEA for African/Neanderthal and

NEADEN for Neanderthal/Denisovan heterozygous states). **c**, Decay curves using the longest 100 Neanderthal segments in each genome. **d**, Estimates of generations between the Neanderthal introgression event and the life of the individual, on the basis of the length of the called segments (*n* = 100). Error bars represent the 95% CI obtained from the chi-squared distribution.

longer than those in other high-coverage ancient hunter-gatherer genomes (Fig. 3c). Three low-coverage Ranis individuals had sufficient data to call segments and yielded similar estimates (union of 95% CIs, 2.7–3.3% for Ranis4, Ranis12 and Ranis87; Supplementary Table 14.4).

Members of the Zlatý kůň/Ranis population co-inhabited Europe with Neanderthals so that recent Neanderthal ancestors in their history are plausible. Previous analysis of greater than 40-kyr-old modern humans from Europe uncovered evidence for such secondary admixtures with Neanderthals[2]. To test whether the Neanderthal ancestry in individuals from the Zlatý kůň/Ranis population fits best to a scenario of a single or multiple introgression events, we fitted the distribution of segments to a single exponential or a mixture of two exponential distributions (Supplementary Tables 13.2 and 14.2). In addition, we applied a dating method to the Ranis13 and Zlatý kůň high-coverage genomes that does not require segments to be called[8] (Supplementary Table 13.1). Both methods found a better fit of the data to a scenario of a single introgression event. Assuming a single-generation pulse of Neanderthal ancestry, the Ranis and Zlatý kůň individuals are estimated to have lived 56–98 generations after the admixture (union of 95% CIs of all applied methods). However, we note that a perhaps more realistic model of continued admixture over multiple generations[34] provides an even better fit and estimates the admixture to have taken place around 80 generations before the individuals lived (Supplementary Table 13.4).

Previous analyses of present-day non-Africans identified five large chromosomal regions that are devoid of Neanderthal ancestry (Neanderthal deserts), and that were probably formed by strong negative selection shortly after admixture[35–37]. The Ranis genomes show no Neanderthal segments that fall in desert regions. However, we confirm the previously reported segment in the Zlatý kůň genome that falls within a desert on chromosome 1 (ref. 3) (Supplementary Information 17). Most of the desert regions were thus established within about 80 generations following the admixture.

Our analyses indicate that the Zlatý kůň/Ranis population split early from the lineage leading to other non-Africans and that they left no descendants among present-day people. The Neanderthal DNA they carry could therefore have been introduced by a separate event from that which introduced the Neanderthal DNA identified in all present-day Out-of-Africa populations. To test whether the introgression events are the same or different, we (1) correlated Neanderthal segments in the high-coverage genomes of Ranis13 and Zlatý kůň with segments in a set of 274 worldwide present-day and 57 ancient modern humans[38,39], and (2) calculated the proportion of edges of Neanderthal segments in the Zlatý kůň or Ranis genomes that are also found in 2,000 present-day and five high-coverage ancient human genomes. Both methods find an excess of correlation or sharing, indicating that the Neanderthal DNA in all ancient and present-day non-Africans, including the Zlatý kůň/Ranis individuals, is best

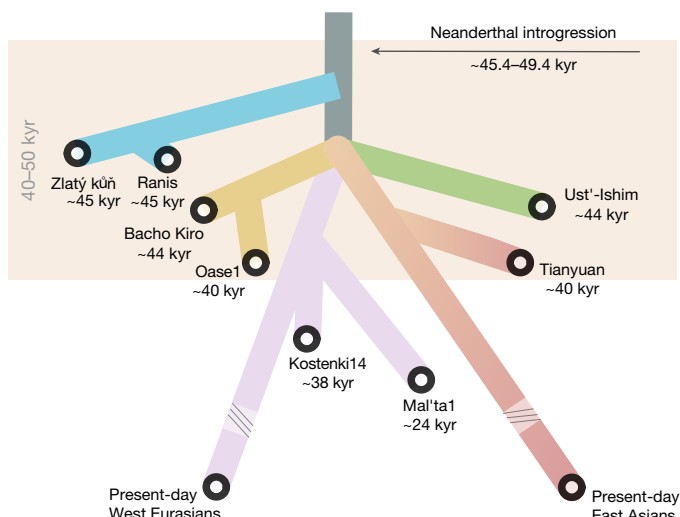

**Fig. 4 | Tree summarizing the main points of the study.** The scheme includes timing of the main Neanderthal introgression event and order of the population separations from the Out-of-Africa lineage.

explained by a shared pulse of Neanderthal introgression (Extended Data Fig. 8).

Because the Ranis individuals carry Neanderthal ancestry from an admixture common to all non-Africans, we can combine the estimated number of generations since this event (56–98 generations), with an assumed generation time of 29 years (refs. 40,41), and the direct radiocarbon date of Ranis13 (43,400–46,580 cal BP)[4] to arrive at a revised date of the common Neanderthal admixture of 45–49 ka (Supplementary Information 13).

## Discussion

In this study, we analysed nuclear genomes from six individuals from Ranis. We found that they were members of a small and isolated population that numbered in the hundreds of individuals per generation. Among these individuals, we identified some with a close degree of biological relatedness, including a mother–daughter pair. Estimates of the number of generations since the Neanderthal admixture consistently fall close to around 80 generations for Ranis individuals (Fig. 3d) and their radiocarbon dates are overlapping, suggesting that these individuals lived close in time to one another. This is in line with the archaeological evidence, such as a low proportion of processed animal remains, limited evidence for the use of fire and a low artefact density at the site which point to short-term occupation of the cave[4,42,43]. We also generated and analysed a high-coverage genome for the Zlatý kůň individual from Czechia, about 230 km from Ranis in Germany. We inferred a fifth- or sixth-degree relationship of this individual to two Ranis individuals and showed that Zlatý kůň falls within the diversity of the Ranis population, in line with a small group size for the Zlatý kůň/Ranis population. Because the Ranis individuals are associated with the LRJ technocomplex, it is plausible that the Zlatý kůň individual may also have been a maker of the LRJ. This association is further supported by several LRJ sites in Czechia, including Nad Kačákem, a cave site about 10 km from Zlatý kůň[44].

In contrast to the genomes from nearly contemporaneous individuals from Bacho Kiro in Bulgaria, the genomes of the Ranis and Zlatý kůň individuals showed no evidence for additional Neanderthal ancestors following the initial admixture. Further mixing with Neanderthals could have been hindered by a short presence and/or the small size of the Zlatý kůň/Ranis population in Europe. However, the Bacho Kiro and Zlatý kůň/Ranis populations could also have differed in the

frequency of Neanderthal encounters along their migratory paths into Europe.

The Zlatý kůň/Ranis population represents the earliest split from the Out-of-Africa population sampled so far, and our results show that this split occurred shortly after a Neanderthal introgression event that took place only around 80 generations before they lived (Fig. 4d). We show that this Neanderthal ancestry originates from the same admixture event that can be detected in all other non-Africans and date this event to 45–49 ka, close to or more recent than most previous estimates of 52–57 ka (ref. 9), 47–65 ka (ref. 45) and 41–54 ka (ref. 8). The narrow confidence interval for our estimate is due to the close proximity of Ranis13 to the admixture event and the precise radiocarbon date of the individual. Because Neanderthal ancestry must have spread among ancestors of all non-Africans, the estimate also provides a date for when a coherent ancestral non-African population must still have existed. This further implies that modern human remains and material cultures older than around 50 kyr found outside of Africa would not represent this non-African population; instead they either resulted from separate Out-of-Africa migrations or they represent populations that split earlier from the ancestors of non-Africans and that were not part of the shared introgression event with Neanderthals. Because all populations that carry ancestry from another archaic lineage, the Denisovans, also carry Neanderthal ancestry from this shared event, we can infer that the Denisovan admixture post-dates 45–49 ka. Further study of ancient genomes, fossils and material cultures will be needed to untangle the events surrounding and following Out-of-Africa migrations, such as the origins of the enigmatic Basal Eurasian lineage, and the earliest waves of modern human movements into Europe and Asia.

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

## Methods

### DNA extraction and sequencing

We generated nuclear DNA data by sequencing 16 existing libraries described in ref. 4 and one in ref. 3, and by sequencing new libraries produced after re-sampling specimens from ref. 4, and three additional specimens. Two of these new specimens, RNI082 and RNI083, are from Ranis and one, ZKU001, is from Zlatý kůň (Supplementary Information 1). Overall, we collected 41 new subsamples from 14 specimens, varying between 37 mg and 3.9 mg in weight, in dedicated clean room facilities at the former Max Planck Institute for the Science of Human History in Jena, Germany, and at the Max Planck Institute for Evolutionary Anthropology (MPI-EVA) in Leipzig, Germany, using a sterile dentistry drill. DNA was extracted following ref. 46 with buffer 'D' and subsequently converted into single-stranded libraries following the automated protocol described in ref. 47. For 25 libraries from 11 specimens we performed single nucleotide polymorphism (SNP) enrichment capture using the 1240k array[15] for population genetics analyses, and for 17 libraries from nine specimens we performed capture using the Archaic Admixture array, specifically Panel 4 (ref. 16), for investigating Neanderthal and Denisovan ancestry. Libraries were captured following the capture protocol presented in ref. 48 and ref. 49. Libraries after 1240k array capture were sequenced on a HiSeq4000 platform, and libraries after Archaic Admixture array capture were sequenced on either a NextSeq500 or a HiSeq4000 platform in the Ancient DNA Core Unit facility at the MPI-EVA. For two of the specimens, RNI010 and RNI087, we performed both complete Y-chromosome capture[50] and Y-mappable capture assay (YMCA)[51] capture and generated sequences on a NextSeq platform. We demultiplexed the resulting sequences on the basis of perfect matching of the expected index combinations, and mapped them to the hg19 reference genome with BWA (v.0.5.10-evan.9-1-g44db244, https://github.com/mpieva/network-aware-bwa) with the ancient parameters ('-n 0.01 -o 2 -l16500')[52] (Supplementary Information 3).

Three libraries from ZKU002 and four libraries from RNI013 were used to obtain the high-coverage Zlatý kůň and Ranis13 genomes, respectively. Sequences from two libraries, ZKU002.A0101 (double-stranded, half-*Escherichia coli* uracil-DNA-glycosylase (UDG) treated) and ZKU002.A0102 (single-stranded, no UDG treatment), were previously published[3]. In this study, ZKU002.A0102 was further sequenced on two full HiSeq4000 runs, using 76-base pair (bp) single-read sequencing. A third library, ZKU002.A0103, was produced from the same DNA extract as the two previous libraries and sequenced at the SciLifeLab in Stockholm, Sweden, on a full Novaseq S4-200 flow-cell using 2 × 75-bp paired-end sequencing. For Ranis13, four single-stranded libraries (RNI013.A0101, RNI013.A0102, RNI013.A0103 and RNI013.A0104) were prepared from the same lysate and pooled together for deeper sequencing. Pooled libraries were sequenced at SciLifeLab following the same set-up as for the ZKU002.A0103 library described above. Adaptors were removed and forward and reverse reads were merged using leeHom (v.1.1.5)[53]. Sequences were assigned to libraries on the basis of the index sequences, allowing for up to one mismatch per index. When libraries were sequenced exclusively on a full lane, all sequences were assigned to a given library, as long as no evidence for extraneous index combinations was found. Assigned sequences were mapped to the human reference genome (*hg19*) with BWA[54] (parameters: -n 0.01 -o 2 -l16500), and duplicates were removed using bam-rmdup (v.0.2, https://github.com/mpieva/biohazard-tools/) separately for each library.

### Data processing, general filters and genotyping

All capture data were filtered for a minimum sequence length of 30 bp and a minimum mapping quality of 25. For the libraries from RNI013 and ZKU002, we used SpAl[55] (v.2) to estimate length cutoffs that ensure a low fraction of spurious alignments. A maximum of 1% spurious alignments was estimated for the length cutoffs of 26, 27 and 27 bp for ZKU002.A0101, ZKU002.A0102 and ZKU002.A0103, respectively. For all four libraries of Ranis13, the length cutoff for at most 1% spurious alignments was 28 bp. We filtered each library accordingly for minimum length, and mapping quality of 25, using SAMtools[56] (v.1.3.1). We also removed alignments with indels for sequences shorter than 35 bp. We applied the genome mappability filter map35_100% as described in ref. 27. To ensure mappability of sequences shorter than 35 bp, we used MapL[55]. We estimated present-day human contamination using AuthentiCT[14] and/or hapCon_ROH[17].

We used snpAD (v.0.3.11)[6,57] to call genotypes for the Zlatý kůň and Ranis13 high-coverage genomes after filtering for a base quality of 30 and a sequence length of 30, and after re-aligning indels using GATK[58] (v.1.3-14). Analyses that are expected to be particularly sensitive to errors used an alternative genotype calling with a 35-bp length cutoff. Error profiles (probabilities of base exchanges) and genotype frequencies were estimated and used to call the most likely genotype at each site, independently for each chromosome. Following previous approaches[6,27,32], we applied further filters to remove sites that fall within the 2.5% extremes of the GC-corrected coverage distribution, and removed tandem repeats[59] and sites called as indels. We also filtered for sites with a minimum depth of 10×, and maximum depth of 50×.

### Detection of biological kinship and inbreeding

We used the software KIN[18] to infer kinship among the specimens with more than 0.05× coverage. We restricted our analyses to on-target sites from the 1240k capture array[15], and provided the model with contamination estimates from AuthentiCT[14] and/or hapCon_ROH[17] as described in Supplementary Information 6. We merged the data from specimens found to be originating from the same individual, and repeated the analysis.

Homozygous regions in the high-coverage genomes of Ranis13 and Zlatý kůň were detected using a previously described method[6,27]. In addition, we detected ROH using 1240k sites in low- and high-coverage genomes with hapROH[28].

### Branch shortening and split time estimates

To estimate the age of Ranis13 and Zlatý kůň, we analysed their high-coverage genomes with two methods: (1) by counting the number of missing mutations in the ancient lineages (branch shortening)[9,27], and (2) by matching demographic histories estimated by the PSMC method[25,32] to estimates for younger individuals.

Branch shortening estimates were based on genotype calls with a minimum sequence length of 35 and we counted only transversion substitutions. The ancestral state at each position was inferred from genome alignments of four apes (chimpanzee, bonobo, gorilla and orangutan—panTro4, bonobo, gorgor3, ponabe2) and differences to the ancestral state were used to estimate branch length. We used the genome of a present-day Mbuti individual (SS6004471, HGDP00982)[27] to calibrate our estimates.

We used PSMC[25] (v.0.6.5) to reconstruct the demographic history of Ranis13 and Zlatý kůň and followed the approach in ref. 32 to estimate the age of both individuals. In brief, after correcting for biases introduced through filtering with the help of simulations (using scrm[60]), we compared the demographic history reconstructed from the genome of a present-day French individual (SS6004468, HGDP00533)[38] with the demographic history reconstructed from the genomes of Ranis13 and Zlatý kůň to estimate the age difference between these two groups. For this, the demographic histories of the younger genome of the French individual were truncated by 0 to 162,429 years in steps of 5,075 years, while simulating ten whole genomes following that truncated demographic history each time. We then filtered this genome identically to Ranis13 and Zlatý kůň, and ran PSMC on it. The truncated histories that best fit the ancient genomes provided an estimate of the time difference to the recent genome.

To estimate the split time between various other Palaeolithic ancient high-coverage genomes and Ranis13/Zlatý kůň, we used momi2 (ref. 21).

## Population genetics

**Using 1240k capture data.** We calculated $f$-statistics using qpDstat from ADMIXTOOLS (https://github.com/DReichLab). The $f_3$-outgroup statistics were used to calculate the affinity matrix with all the possible hunter-gatherer pairwise combinations. The $f_4$-statistics were used to test for cladality and admixture. For all plots, standard errors were calculated with the default block jackknife and we consistently display 3 standard errors.

Multidimensional scaling analysis was performed using the R implementation cmdscale. Euclidean distances were computed with the genetic distances calculated from the $f_3$-outgroup matrix on the form $1 - f_3$ pairwise values among all possible hunter-gatherer pairwise combinations. We used qpGraph from ADMIXTOOLS (https://github.com/DReichLab) to determine the phylogenetic positions of the Ranis and Zlatý kůň individuals, following the tree structure reported in ref. 2 and ref. 17.

**Using high-coverage shotgun data.** We calculated $D$-statistics using the genotype calls (snpAD[6,57] v.0.3.11) from the high-coverage genomes of Zlatý kůň and Ranis13, along with various ancient and present-day human genomes, as well as the Neanderthal and Denisovan high-quality genomes (Supplementary Information 9). A random allele was chosen per individual at heterozygous calls, and subsetted to the biallelic sites. To infer the state of the ape-ancestor outgroup, the aligning base in the chimpanzee, bonobo, gorilla and orangutan genomes had to match. $D$-statistics were computed as $\frac{BABA - ABBA}{ABBA + BABA}$ and confidence intervals were estimated on the basis of a weighted block jackknife procedure in 5-Mb windows, and plotted with 3 standard errors (refs. 3,6,20,61).

## Neanderthal ancestry

We used admixfrog[33] (v.0.7.1) to infer Neanderthal ancestry segments in the high-coverage Zlatý kůň and Ranis13 genomes, as well as from the low-coverage genome data obtained through Archaic Admixture capture[16]. We built our reference panels from allele information of the high-coverage genomes of Denisova 3 (ref. 52), Vindija 33.19 (ref. 6), Denisova 5 (or the Altai Neanderthal)[27], Chagyrskaya 8 (ref. 32), the chimpanzee reference genome (panTro6) and all the female Mende, Mandenka, Yoruba, Esan and Luhya individuals from the 1000 Genomes Project at the sites ascertained by the Archaic Admixture array (Fu et al., 2015, Panel 4)[16] and, for the X-chromosome sites, from the Extended Archaic ascertainment[39]. The genetic distances were assigned using either the African American recombination map[62] or the deCODE map[63].

The number of introgression events and the timing of these events were inferred by the Moorjani et al.[8] dating method on the genotype calls, and another method that fits Neanderthal segment lengths to a single exponential distribution, or a mixture of two exponential distributions, corresponding to single or two admixture events, respectively. A length cutoff of 0.2 cM was applied for the second method to avoid shorter segments that could be generated by incomplete lineage sorting (Supplementary Information 13). In addition to a simple pulse model, we also tried to fit an extended pulse as described in ref. 34. We quantified the Neanderthal ancestry in the genomes of Ranis individuals and Zlatý kůň on the basis of the segments detected by admixfrog following ref. 64, and by applying a 'direct $f_4$-ratio test'[20,65]. We also overlapped Neanderthal segments with the intersections of the previously reported Neanderthal and Denisovan desert regions using BEDTools[35–37,66].

We investigated whether Neanderthal segments in the genomes of Ranis13 and Zlatý kůň are shared with later individuals either by correlating the location of introgressed segments[39] (Supplementary Information 15) or by testing for overlap of the edges of Neanderthal segments (Supplementary Information 16).

## Y-chromosome and phenotypes

Y-SNP lists from the ISOGG[67] (International Society of Genetic Genealogy) collection (v.15.73) were investigated after filtering $C \rightarrow T$ and $G \rightarrow A$ substitutions on the forward and reverse strands, respectively. We then manually called the most resolved Y haplogroup for which we have derived calls, and minimal ancestral calls (Supplementary Information 18).

To infer HLA haplotypes at the five most polymorphic HLA loci (HLA-A, -B, -C, -DRB1 and -DQB1), we followed ref. 68 (Supplementary Information 19). Merged fastq files were filtered to remove reads shorter than 30 bp. Sequence data were aligned to an HLA reference file using Bowtie2 (ref. 69) (v.2.2.6) in local alignment mode. Alignments were manually inspected to reconstruct, at each locus, the consensus sequences of the two haplotypes while considering only unambiguous alignments to the locus of interest. Consensus sequences were compared with known four-digit HLA alleles to find the best-matching sequences and thus define allele calls. Allele calls at the class I loci were further corroborated using OptiType[70] (v.1.3.3).

We investigated 43 variants that are associated with phenotypic variation, including lactose persistence and pigmentation (Supplementary Information 20). We did this by counting the number of sequences carrying the effect allele versus the non-effect allele for each variant in all genomes with ≥1× coverage on the 1240k sites (high-coverage shotgun data from Ranis13 and Zlatý kůň, and low-coverage capture data from Ranis4, Ranis12 and Ranis87).

We used Rstudio (v.2022.12.0+353, http://www.rstudio.com/) and the following packages for data visualization: cowplot (v.1.1.2), ggplot2 (v.3.4.2, https://ggplot2.tidyverse.org), tidyr (v.1.3.0, https://github.com/tidyverse/tidyr), dplyr (v.1.1.4, https://github.com/tidyverse/dplyr), magrittr (v.2.0.3, https://github.com/tidyverse/magrittr), scales (v.1.3.0, https://github.com/r-lib/scales) and MetBrewer (v.0.2.0, https://github.com/BlakeRMills/MetBrewer).

## Reporting summary

Further information on research design is available in the Nature Portfolio Reporting Summary linked to this article.

## Data availability

All newly reported ancient nuclear DNA data are archived in the European Nucleotide Archive (accession no. PRJEB78725). Additionally, the alignment files used for the HLA analyses can be accessed through https://doi.org/10.17617/3.GHAALO.

## Code availability

We provide the code for computing $D$-statistics and the Neanderthal ancestry breakpoint sharing analysis through the Max Planck Digital Library (https://doi.org/10.17617/3.EGKV28, https://doi.org/10.17617/3.0VLEOH and https://doi.org/10.17617/3.TG4TO4), and for running the PSMC analyses through GitHub (https://github.com/StephanePeyregne/calibratePSMC).

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

**Acknowledgements** We thank A. Aximu-Petri, S. Nagel, A. Busch, L. Gerullat, A. Weihmann, B. Schellbach, J. Visagie and T. Lamnidis for assistance in the laboratory, sequencing and data processing; R. Radzeviciute for handling and transferring samples; S. Tüpke for professional photographs and assistance with some of the figures; H. Temming for micro-CT scans of the specimens; S. Zhang for providing the reuse permit for the Tianyuan specimen photograph; and L. Wacker for running the AMS for radiocarbon dating. We thank S. Schiffels, L. Pagani and L. Vallini for helpful discussions and L. Huang for help with processing of HGDP/SGDP data. We acknowledge support from the National Genomics Infrastructure in Stockholm funded by the Science for Life Laboratory, the Knut and Alice Wallenberg Foundation and the Swedish Research Council, and NAISS for assistance with massively parallel sequencing and access to the UPPMAX computational infrastructure. This study was funded by the Max Planck Society. H. Rougier received funding from CSUN's RSCA Awards and College of Social and Behavioral Sciences. V.V.-M. is supported by the grant 'Ayudas para contratos Ramón y Cajal' (no. RYC2022-035700-I) funded by Ministerio de Ciencia, Innovación y Universidades. L.N.M.I. and B.M.P. are funded by the European Research Council (ERC) under the European Union's Horizon Europe research and innovation programme (grant agreement no. 101042421 NEADMIX, awarded to B.M.P.). E.I.Z. is supported by the Miller Institute for Basic Research in Science, University of California Berkeley. V.S.-M. is supported by a Fyssen Foundation postdoctoral fellowship (2023–2025). P.V. and J.B. are supported by The Czech Science Foundation grant no. GA23-06822S. P.V. is also supported by The Ministry of Culture of the Czech Republic (DKRVO grant no. 2024-2028/7.I.a, 00023272). This project has received funding from the ERC under the European Union's Horizon 2020 research and innovation programme (grant agreement no. 948365, awarded to F.W.), and ERC Starting grant credited to K.I.B. under grant agreement no. 805268 (CoDisEASe).

**Author contributions** This study was designed by A.P.S., H. Rougier, K.P. and J. Krause. H. Rougier, D.M., G.M.S., K.R., V.S.-M., A. Stoessel, H.D., J.O., H.Z., F.W., M.W., S.P.M., T.S., J.-J.H., P.V., J.B. and H.M. participated in archaeological excavation, provided anthropological assessment or contributed to the curation of skeletal materials. Sample preparations and laboratory work were conducted by A.P.S., E.E., A.F., H.F., E.I.Z., R.A.B., A. Schmidt, J.Z., B.N., A.P. and C.P., and subsequent analyses were performed by A.P.S., V.V.-M., Y.H., L.N.M.I., A.B.M., S.P., C.d.F., A.B.R., F.P., F.M., H.F and M.H., with supervision from J. Kelso, K.I.B., B.M.P., M.M., H. Ringbauer and K.P. The original manuscript was prepared by A.P.S., K.P. and J. Krause, with contributions from all co-authors.

**Funding** Open access funding provided by Max Planck Society.

**Competing interests** The authors declare no competing interests.

**Additional information**
**Correspondence and requests for materials** should be addressed to Arev P. Sümer, Kay Prüfer or Johannes Krause.

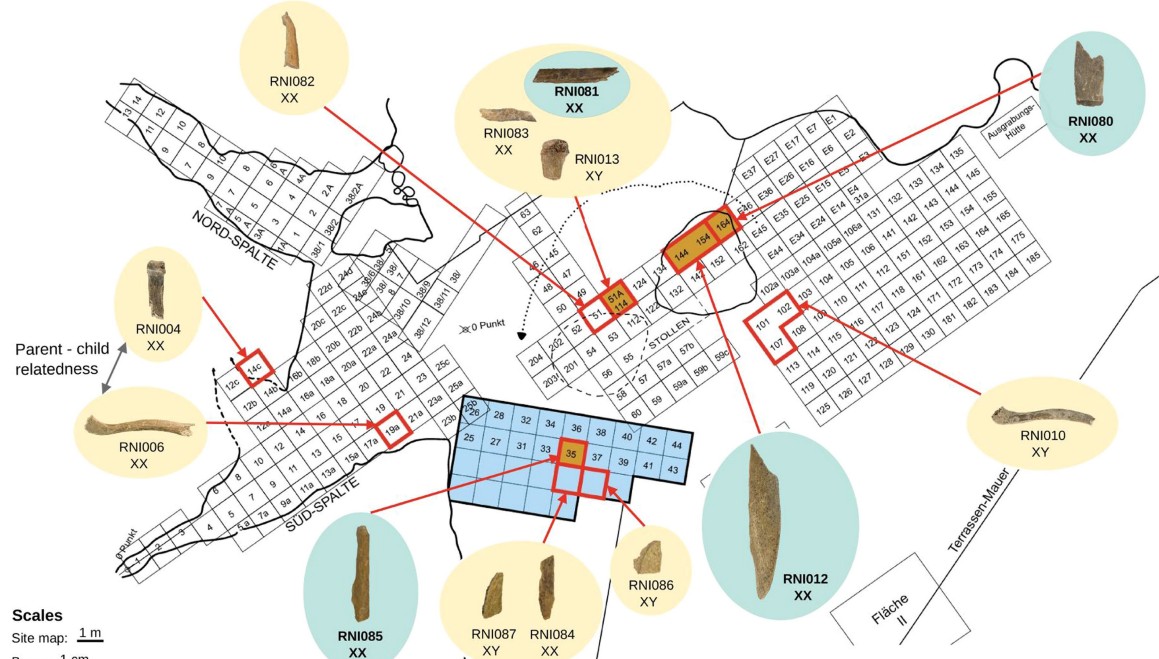

**Extended Data Fig. 1 | Map of Ranis with the location of the specimens (named RNI0XX) included in this study.** Green circles contain specimens from the same individual, Ranis12, and the dark orange areas are the squares they were excavated from. Genetic sexes of the individuals that the specimens belonged to are shown below their IDs. The blue area indicates the squares excavated in 2016–2022 while the rest of the grid was excavated in 1932–1938. Credit: background raster is adapted from ref. 4, Springer Nature Limited, under a Creative Commons licence, CC BY 4.0.

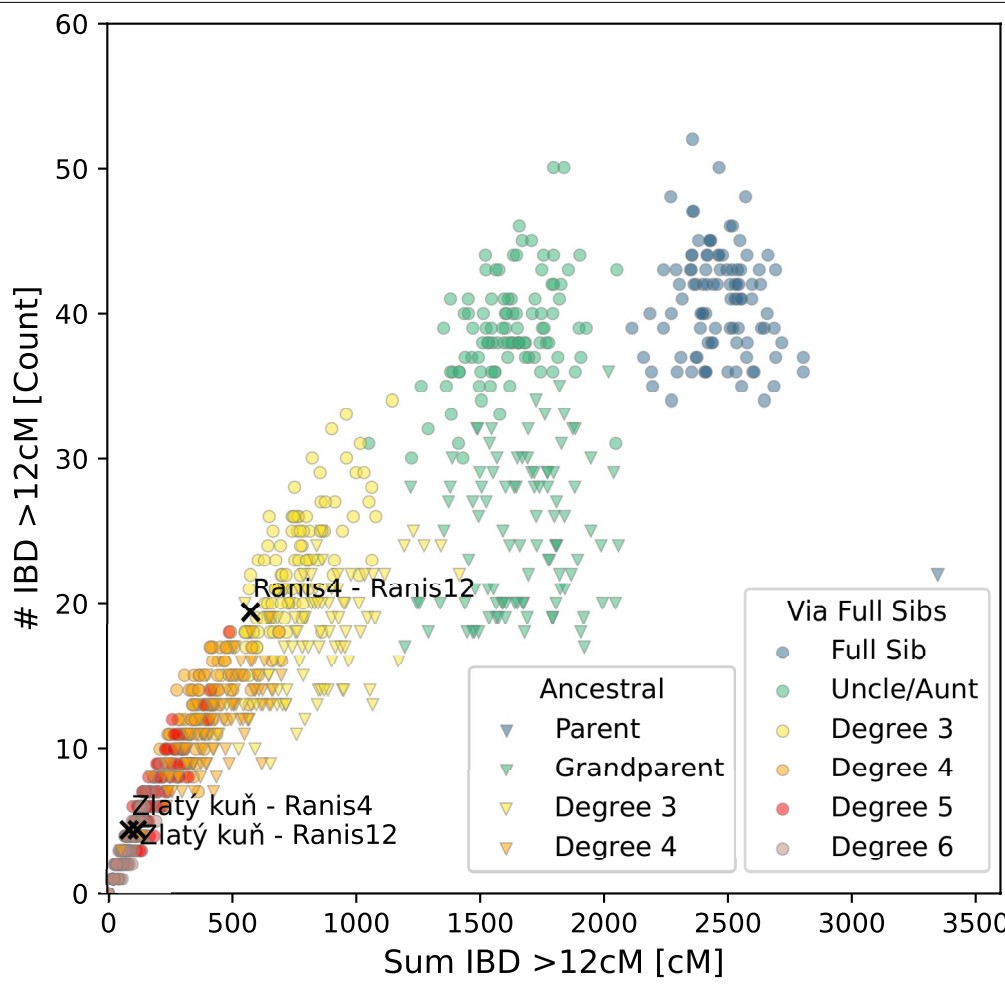

**Extended Data Fig. 2 | Number of identical by descent (IBD) segment sharing for the three pairs with the highest values among the Ranis and Zlatý kůň individuals, plotted against simulated relatives of various degrees.** Values on the x-axis stand for the sum of the shared IBD segments longer than 12 cM and values on the y-axis are the numbers of IBD segments longer than 12 cM. Biological relatives are simulated with Ped-sim[26]. Units are reported in square brackets in the axis labels. For the Ranis and Zlatý kůň pairs, we subtracted the expected background IBD sharing for a homogeneous population of effective size 299.7 (estimated from ROH blocks, Supplementary Informations 7 & 8) before plotting it on top of simulated pairs. Ranis6 is not included due to low data quality.

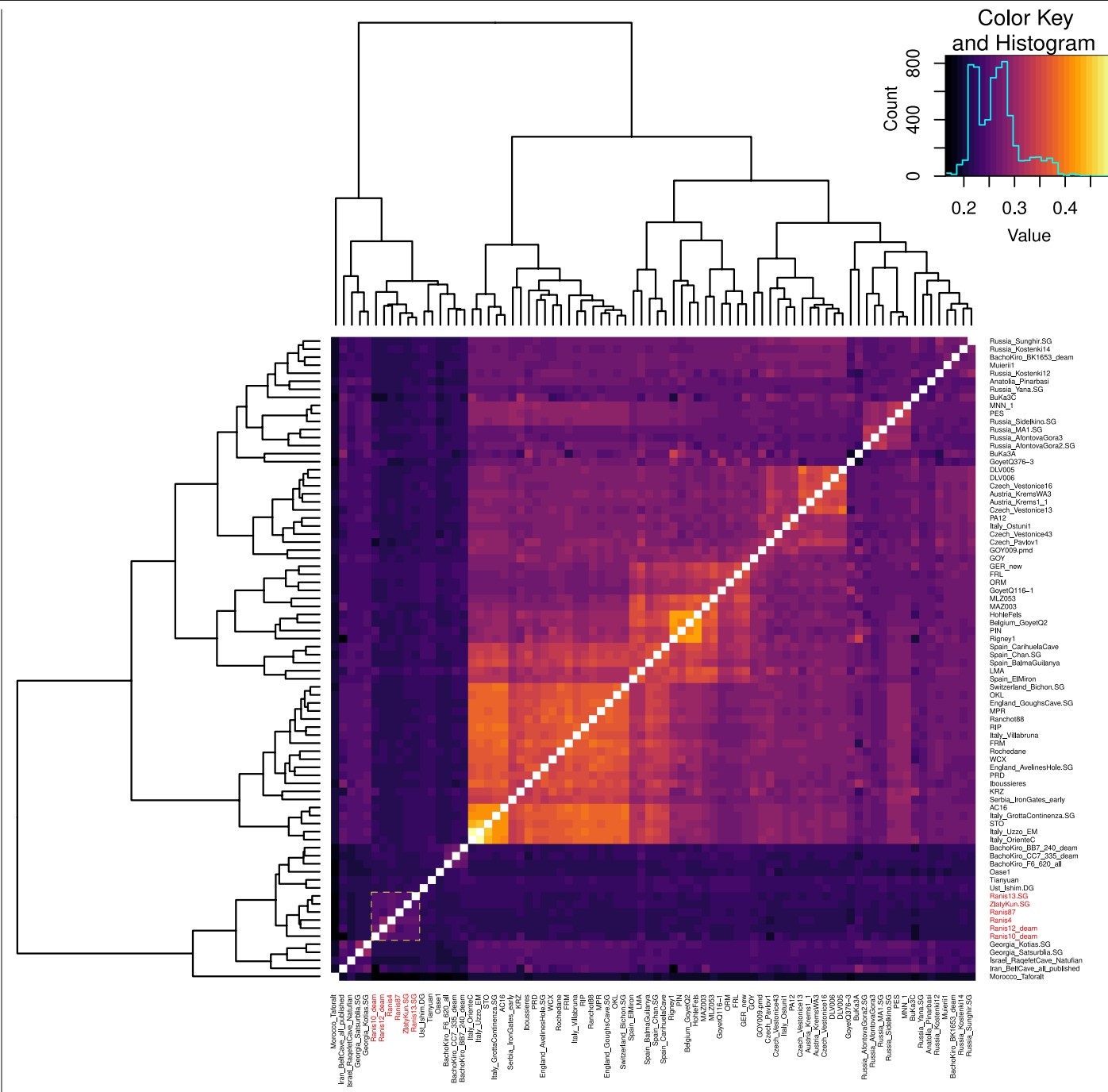

**Extended Data Fig. 3 | Heatmap of pairwise comparisons of hunter-gatherers in *f3*-outgroup statistics.** An Mbuti genome is used as the outgroup.

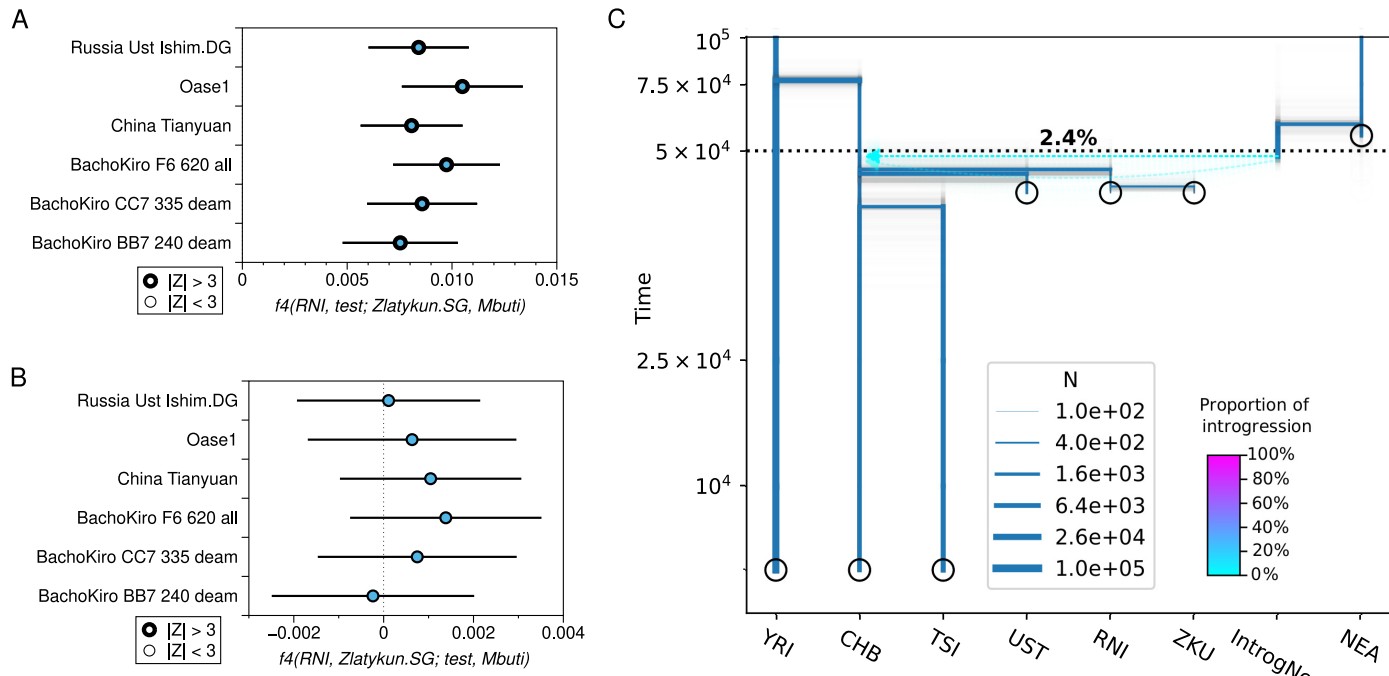

**Extended Data Fig. 4 | Population relationships. A.** & **B.** *F*-statistics investigating the relationship between the Ranis and Zlatý kůň individuals and the other early modern humans using the 1240k array sites for Ranis as a group and the high-coverage genome of Zlatý kůň. Error bars correspond to three standard errors. **C.** Tree inferred by momi2, using high-coverage genomes. We use the following abbreviations: Yoruba (YRI), Han Chinese (CHB), Tuscani (TSI), Ust'-Ishim (UST), Ranis13 (RNI), Zlatý kůň (ZKU), and Vindija33.19 (NEA). IntrogNea represents the Neanderthal lineage that introgressed into the modern human population.

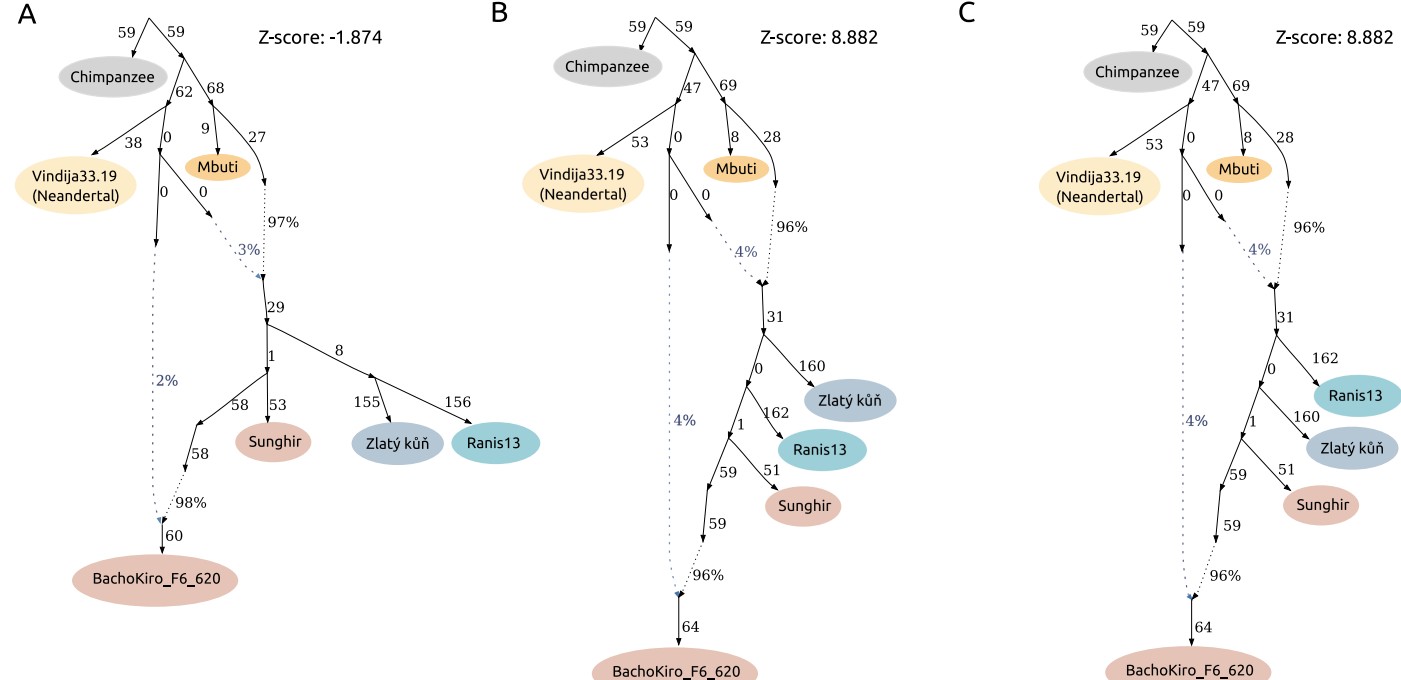

**Extended Data Fig. 5 | qpGraphs.** Analyses were conducted following the tree structure reported in Hajdinjak et al.[2], and Posth et al.[17]. **A.** Ranis13 and Zlatý kůň modeled as sister clades, **B.** Zlatý kůň representing an earlier split, and **C.** Ranis13 representing an earlier split from the Out-of-Africa lineage.

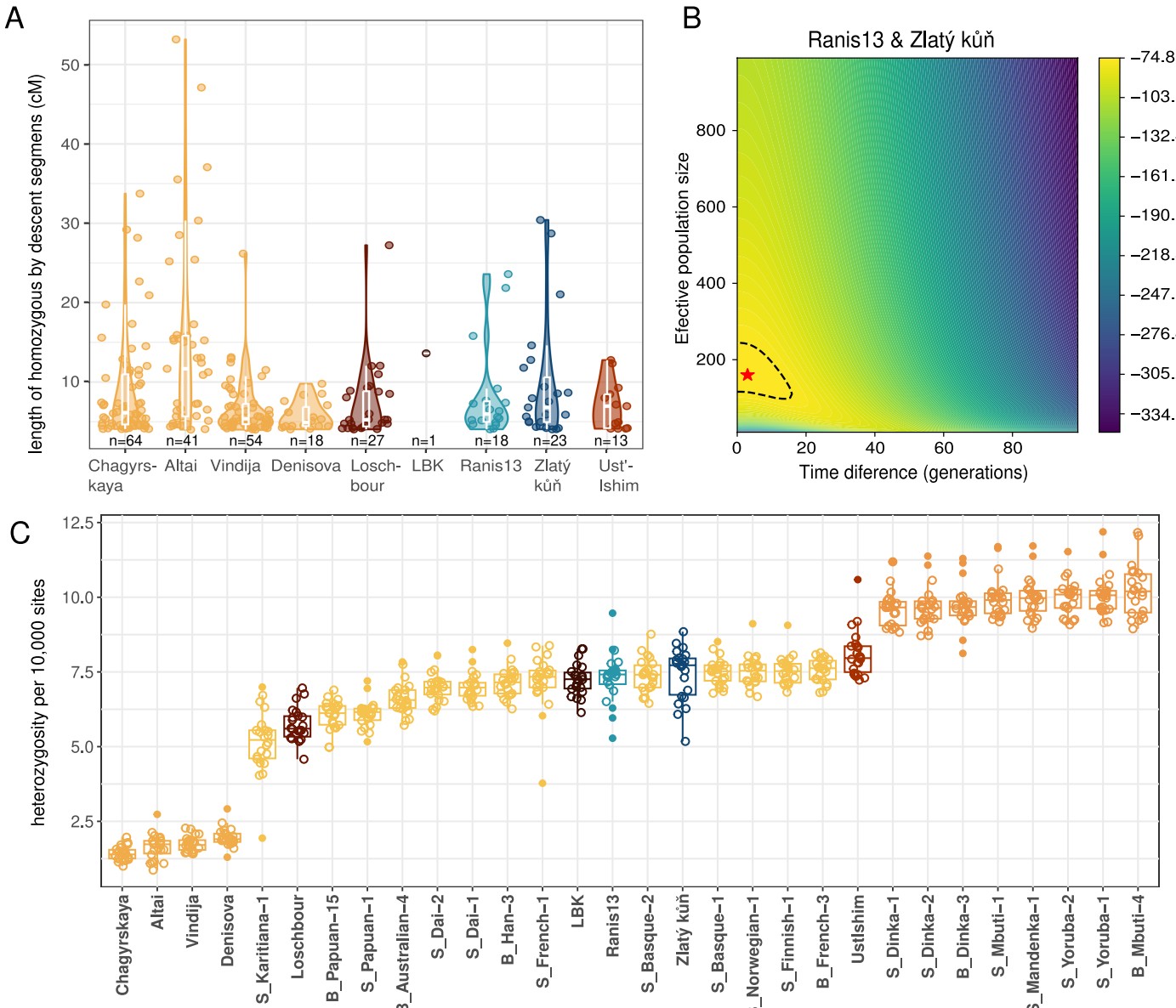

**Extended Data Fig. 6 | Inbreeding, population size and heterozygosity estimates. A**. Violin plots indicating the length distributions of the segments that are homozygous by descent (HBD, or ROH), using the African-American recombination map. The width of the violins is proportional to the number of HBD segments. **B**. Likelihood of IBD shared between Ranis13 and Zlatý kůň over a 2D grid of effective population size and the absolute value of time difference. The red star indicates the maximum likelihood estimate and the black dashed circle indicates the 95% confidence region. The colour bar indicates the log-likelihood values. **C**. Average heterozygosity (x10,000) in the autosomes of high-coverage ancient and present-day human genomes. Each point represents the heterozygosity estimate on a single autosome (n = 22). Archaics are in light orange and Africans are in dark orange, while non-Africans are in yellow. Other colours uniquely represent ancient modern humans. The lower and upper hinges in the boxplots in **A** and **C** correspond to the 25th and 75th percentiles. Upper and lower whiskers are at most 1.5 inter-quartile range away from the upper and lower hinges, respectively.

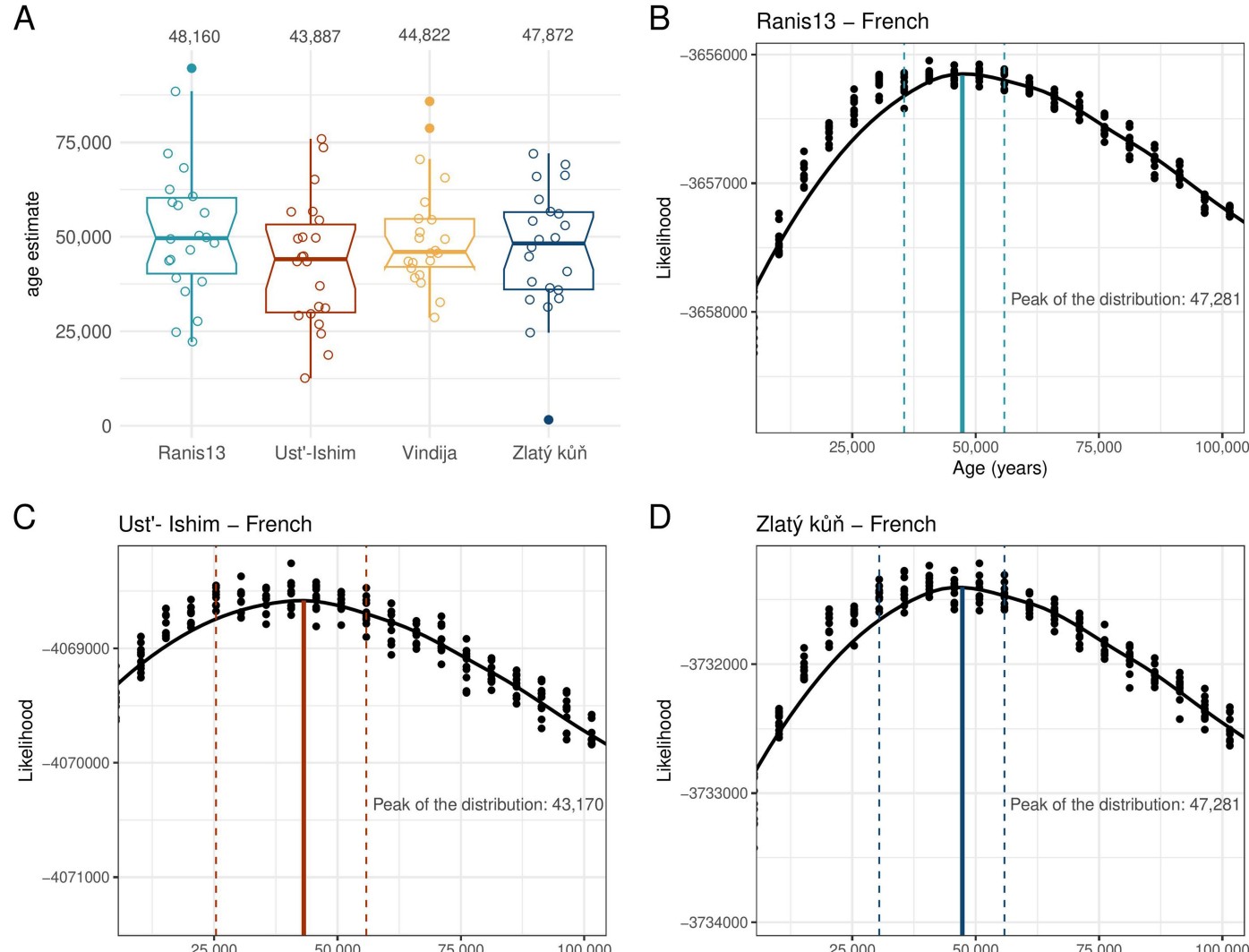

**Extended Data Fig. 7 | Molecular age estimates.** These ages were obtained (**A.**) based on branch shortening by missing mutation counts, with each point representing a chromosome and value on the top being the weighted mean value excluding the outliers (filled points) and (**B.-D.**) using the PSMC curves for three high-coverage early modern humans, in comparison to the genome of a present-day French individual. The lower and upper hinges in the boxplots in **A** correspond to the 25th and 75th percentiles. Upper and lower whiskers are at most 1.5 inter-quartile range away from the upper and lower hinges, respectively.

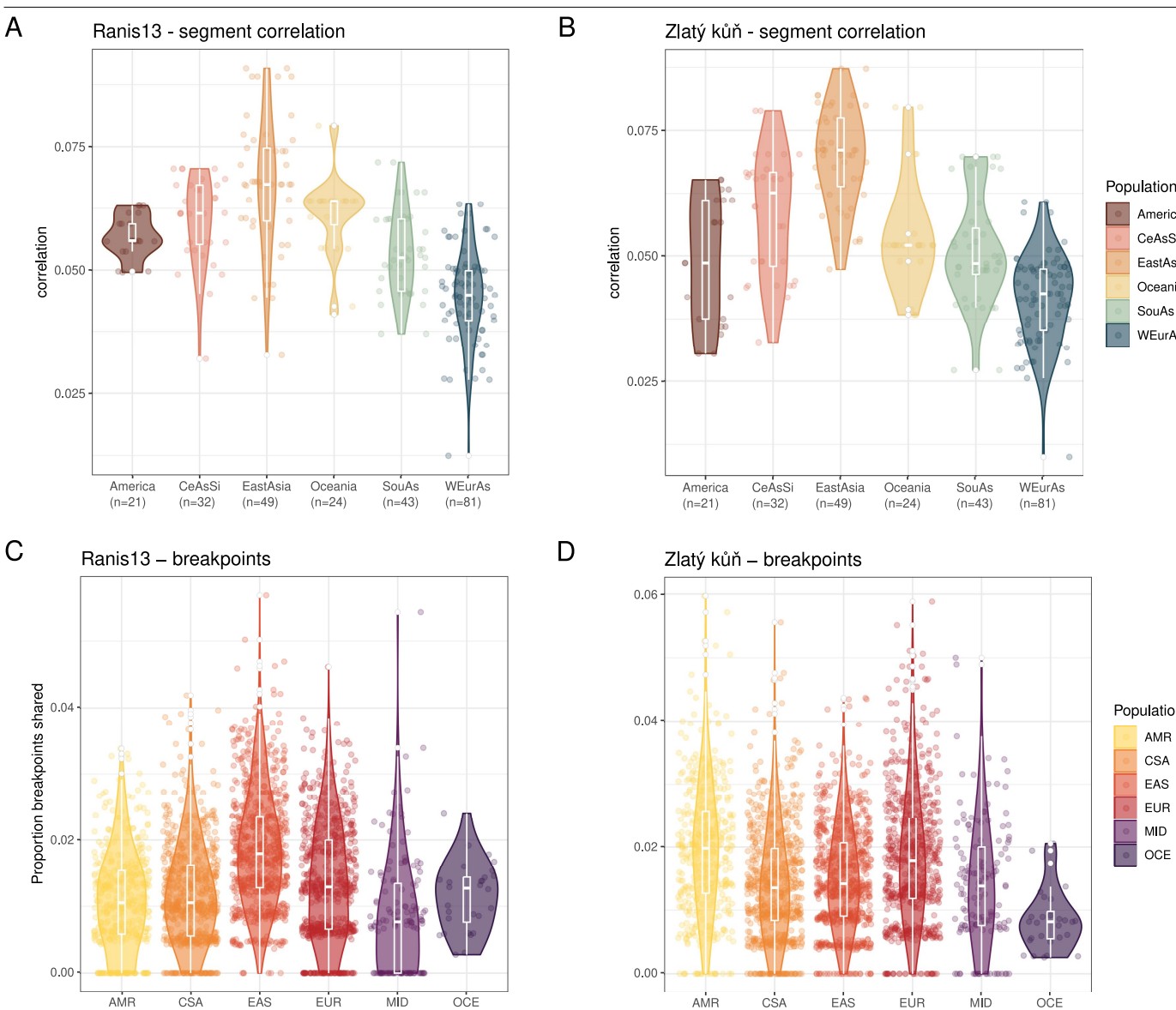

**Extended Data Fig. 8 | Sharing of the Neanderthal ancestry with present-day populations.** Correlations of the Neanderthal segments with the present-day SGDP populations for (**A**.) Ranis13 and (**B**.) Zlatý kůň. Each point represents a genome in the corresponding SGDP super-population. Correlation of the recombination breakpoints with HGDP and 1000 genomes populations for (**C**.) Ranis13 and (**D**.) Zlatý kůň. We use the following abbreviations: Americans (AMR), Central South Asians (CSA), East Asians (EAS), Europeans (EUR), Middle Easterners (MID), and Oceanians (OCE), and each point represents a genome from these populations. The lower and upper hinges in the boxplots in white correspond to the 25th and 75th percentiles. Upper and lower whiskers are at most 1.5 inter-quartile range away from the upper and lower hinges, respectively.

Kay Prüfer
Johannes Krause

# Reporting Summary

## Statistics

For all statistical analyses, confirm that the following items are present in the figure legend, table legend, main text, or Methods section.

| n/a | Confirmed | |
|---|---|---|
| ☐ | ☒ | The exact sample size (*n*) for each experimental group/condition, given as a discrete number and unit of measurement |
| ☐ | ☒ | A statement on whether measurements were taken from distinct samples or whether the same sample was measured repeatedly |
| ☐ | ☒ | The statistical test(s) used AND whether they are one- or two-sided<br>*Only common tests should be described solely by name; describe more complex techniques in the Methods section.* |
| ☒ | ☐ | A description of all covariates tested |
| ☐ | ☒ | A description of any assumptions or corrections, such as tests of normality and adjustment for multiple comparisons |
| ☐ | ☒ | A full description of the statistical parameters including central tendency (e.g. means) or other basic estimates (e.g. regression coefficient) AND variation (e.g. standard deviation) or associated estimates of uncertainty (e.g. confidence intervals) |
| ☐ | ☒ | For null hypothesis testing, the test statistic (e.g. *F*, *t*, *r*) with confidence intervals, effect sizes, degrees of freedom and *P* value noted<br>*Give P values as exact values whenever suitable.* |
| ☐ | ☒ | For Bayesian analysis, information on the choice of priors and Markov chain Monte Carlo settings |
| ☒ | ☐ | For hierarchical and complex designs, identification of the appropriate level for tests and full reporting of outcomes |
| ☒ | ☐ | Estimates of effect sizes (e.g. Cohen's *d*, Pearson's *r*), indicating how they were calculated |

*Our web collection on statistics for biologists contains articles on many of the points above.*

## Software and code

Policy information about availability of computer code

| Data collection | no software was used |
|---|---|
| Data analysis | BWA (v. 0.5.10-evan.9-1-g44db244, https://github.com/mpieva/network-aware-bwa) , leeHom (v1.1.5), bam-rmdup (v0.2), SAMtools (v. 1.3.1), MapL (v0.1), AuthentiCT, hapCon_ROH, snpAD (v. 0.3.11), GATK (v. 1.3-14), SpAl (v. 2) , KIN, PSMC (v. 0.6.5), scrm (v. 1.7.3), qp3Pop (v.435; ADMIXTOOLS v.5.1), qpDstat (v.755; ADMIXTOOLS v.5.1), qpAdm (v.810), admixfrog (v.0.7.1), admixtools (v.2.0.0), BEDTools (v.2.25.0), Bowtie (v. 2.2.6), OptiType (v. 1.3.3). Data visualization was performed in RStudio (v.2022.12.0+353). The following R packages were used for visualization: cowplot (v.1.1.2), ggplot (v.3.4.2), tidyr (v.1.3.0), dplyr (v.1.1.4), magrittr (v.2.0.3), scales (v.1.3.0), MetBrewer (v.0.2.0). |

For manuscripts utilizing custom algorithms or software that are central to the research but not yet described in published literature, software must be made available to editors and reviewers. We strongly encourage code deposition in a community repository (e.g. GitHub). See the Nature Portfolio guidelines for submitting code & software for further information.

## Data

Policy information about availability of data

All manuscripts must include a data availability statement. This statement should provide the following information, where applicable:
- Accession codes, unique identifiers, or web links for publicly available datasets
- A description of any restrictions on data availability
- For clinical datasets or third party data, please ensure that the statement adheres to our policy

All newly reported ancient nuclear DNA data is archived in the European Nucleotide Archive (accession no. PRJEB78725). Additionally, the Alignment files used for the HLA analyses can be accessed through https://doi.org/10.17617/3.GHAALO.

Publicly available data was obtained from the following sources:
- Publicly available data in the European Nucleotide Archive: PRJEB39134 (Hajdinjak et al., 2021), PRJEB64496 (Bennett et al. 2023), PRJEB51862 (Posth et al., 2023), PRJEB58642 (Villalba-Mouco et al., 2023), PRJEB21157 (Prüfer et al., 2017), ERP002097 (Prüfer et al., 2014),
- Publicly available data in http://ftp.eva.mpg.de/genomes/ for high-coverage Loschbour, Stuttgart (LBK), Ust'-Ishim, Iceman and Beethoven genomes. Similarly, the high-coverage Neandertal genome from Chagyrskaya8 is publicly available in http://ftp.eva.mpg.de/neandertal/Chagyrskaya.
- High-coverage genome from Denisova3 is available in Short Read Archive: SRA047577(Meyer et al. 2012).
- SGDP data is available in the European Nucleotide Archive: PRJEB9586 and ERP010710 (Mallick et al., 2016). HGDP data is available in European Nucleotide Archive: PRJEB6463 and PRJEB14173 (Bergström et al., 2020).
- The HGDP and 1000 genomes curated data used for the Neandertal breakpoint sharing analyses is available through https://gnomad.broadinstitute.org/downloads#v3-hgdp-1kg
- hg19 reference genome is available through https://www.ncbi.nlm.nih.gov/datasets/genome/GCF_000001405.13/, and 1000 genome data can be obtained at https://www.internationalgenome.org/data/.

## Research involving human participants, their data, or biological material

Policy information about studies with human participants or human data. See also policy information about sex, gender (identity/presentation), and sexual orientation and race, ethnicity and racism.

| | |
|---|---|
| Reporting on sex and gender | We use biological data (genomes) from individuals that lived ~45,000 years ago, and infer their biological sex using the genomic data. We do not infer gender. |
| Reporting on race, ethnicity, or other socially relevant groupings | n/a |
| Population characteristics | n/a |
| Recruitment | n/a |
| Ethics oversight | n/a |

Note that full information on the approval of the study protocol must also be provided in the manuscript.

# Field-specific reporting

Please select the one below that is the best fit for your research. If you are not sure, read the appropriate sections before making your selection.

☒ Life sciences ☐ Behavioural & social sciences ☐ Ecological, evolutionary & environmental sciences

For a reference copy of the document with all sections, see nature.com/documents/nr-reporting-summary-flat.pdf

# Life sciences study design

All studies must disclose on these points even when the disclosure is negative.

| | |
|---|---|
| Sample size | Our sample consists of thirteen early modern human specimens from Ilsenhöhle in Ranis, Germany, described in Mylopotamitaki et al., 2024 in Nature. In addition, we generate additional data from the Zlaty kun individual from Czechia, introduced in Prüfer et al., 2021 in Nat. Ecol. Evol., and another specimen from the same site. |
| Data exclusions | Data from five specimens were excluded (RNI082, RNI083, RNI084, RNI086 and ZKU001) due to low endogenous DNA content in the librries obtained from these specimens, and/or high levels of present-day human DNA contamination. In addition, all sequences shorter than 30 basepairs and with a mapping quality of less than 25 were excluded. |
| Replication | Re sampled multiple individuals from the same site and because they belong to the same population, could replicate the popultion genetics results. Other than this, replication is not applicable. |
| Randomization | Randomization is not relevant and not carried out for this study. This is because we start the study with the specimens in hand and analyze |

| Randomization | the data from these specimens. We focus on the sequences per individual, independently and as a population. Randomization is not applicable in this context. |
|---|---|
| Blinding | Blinding is not applicable for ancient DNA/population genetics studies. We do not compare different treatments on different groups. |

# Behavioural & social sciences study design

All studies must disclose on these points even when the disclosure is negative.

| Study description | |
|---|---|
| Research sample | |
| Sampling strategy | |
| Data collection | |
| Timing | |
| Data exclusions | |
| Non-participation | |
| Randomization | |

# Ecological, evolutionary & environmental sciences study design

All studies must disclose on these points even when the disclosure is negative.

| Study description | |
|---|---|
| Research sample | |
| Sampling strategy | |
| Data collection | |
| Timing and spatial scale | |
| Data exclusions | |
| Reproducibility | |
| Randomization | |
| Blinding | |

Did the study involve field work? ☐ Yes  ☒ No

## Field work, collection and transport

| Field conditions | No field work was done for this study. |
|---|---|
| Location | |
| Access & import/export | |
| Disturbance | |

# Reporting for specific materials, systems and methods

We require information from authors about some types of materials, experimental systems and methods used in many studies. Here, indicate whether each material, system or method listed is relevant to your study. If you are not sure if a list item applies to your research, read the appropriate section before selecting a response.

## Materials & experimental systems

| n/a | Involved in the study |
|-----|-----------------------|
| ☒ | ☐ Antibodies |
| ☒ | ☐ Eukaryotic cell lines |
| ☐ | ☒ Palaeontology and archaeology |
| ☒ | ☐ Animals and other organisms |
| ☒ | ☐ Clinical data |
| ☒ | ☐ Dual use research of concern |
| ☒ | ☐ Plants |

## Methods

| n/a | Involved in the study |
|-----|-----------------------|
| ☒ | ☐ ChIP-seq |
| ☒ | ☐ Flow cytometry |
| ☒ | ☐ MRI-based neuroimaging |

## Antibodies

| Antibodies used | |
|---|---|
| Validation | |

## Eukaryotic cell lines

Policy information about cell lines and Sex and Gender in Research

| Cell line source(s) | |
|---|---|
| Authentication | |
| Mycoplasma contamination | |
| Commonly misidentified lines (See ICLAC register) | |

## Palaeontology and Archaeology

| Specimen provenance | Specimens were collected from Ilsenhöhle in Ranis, Germany and Zlaty kun in Czechia. Permits were provided in Mylopotamitaki et al., 2024 in Nature and Prüfer et al., 2021 in Nat. Ecol. Evol.. |
|---|---|
| Specimen deposition | Specimens were either returned to their corresponding collections (National Museum, Prague, Czechia, Landesamt für Denkmalpflege und Archäologie Sachsen-Anhalt-Landesmuseum für Vorgeschichte, Halle, Germany, Thuringian State Office for the Preservation of Historical Monuments and Archaeology, Weimar, Germany) or kept for further sampling in the cleanroom facilities of the Max Planck Institute for Evolutionary Anthropology in Leipzig. |
| Dating methods | Collagen was extracted following the acid-base-acid plus ultrafiltration protocol for small samples outlined in Fewlass et al., 2019[13] in the Ancient Genomics Lab at the Francis Crick Institute, UK. The extracted collagen was graphitised on an AGE 3 system[14] and dated on a MICADAS accelerator mass spectrometer (AMS)[15,16] at the Laboratory for Ion Beam Physics at ETH Zurich, Switzerland. Dates were calibrated using the IntCal20 calibration curve in OxCal 4.4. |

☒ Tick this box to confirm that the raw and calibrated dates are available in the paper or in Supplementary Information.

| Ethics oversight | No ethical guidance was required. Sampling was carried out under the guidance and with the permission of archaeologists/curators responsible for the material. |
|---|---|

Note that full information on the approval of the study protocol must also be provided in the manuscript.

## Animals and other research organisms

Policy information about studies involving animals; ARRIVE guidelines recommended for reporting animal research, and Sex and Gender in Research

| Laboratory animals | |
|---|---|
| Wild animals | |
| Reporting on sex | |

| Field-collected samples | |
| --- | --- |
| Ethics oversight | |

Note that full information on the approval of the study protocol must also be provided in the manuscript.

# Clinical data

Policy information about clinical studies
All manuscripts should comply with the ICMJE guidelines for publication of clinical research and a completed CONSORT checklist must be included with all submissions.

| Clinical trial registration | |
| --- | --- |
| Study protocol | |
| Data collection | |
| Outcomes | |

# Dual use research of concern

Policy information about dual use research of concern

## Hazards

Could the accidental, deliberate or reckless misuse of agents or technologies generated in the work, or the application of information presented in the manuscript, pose a threat to:

No | Yes

☐ ☐ Public health

☐ ☐ National security

☐ ☐ Crops and/or livestock

☐ ☐ Ecosystems

☐ ☐ Any other significant area

## Experiments of concern

Does the work involve any of these experiments of concern:

No | Yes

☐ ☐ Demonstrate how to render a vaccine ineffective

☐ ☐ Confer resistance to therapeutically useful antibiotics or antiviral agents

☐ ☐ Enhance the virulence of a pathogen or render a nonpathogen virulent

☐ ☐ Increase transmissibility of a pathogen

☐ ☐ Alter the host range of a pathogen

☐ ☐ Enable evasion of diagnostic/detection modalities

☐ ☐ Enable the weaponization of a biological agent or toxin

☐ ☐ Any other potentially harmful combination of experiments and agents

# Plants

| Seed stocks | |
| --- | --- |
| Novel plant genotypes | |
| Authentication | |

# ChIP-seq

## Data deposition

☐ Confirm that both raw and final processed data have been deposited in a public database such as GEO.

☐ Confirm that you have deposited or provided access to graph files (e.g. BED files) for the called peaks.

Data access links
*May remain private before publication.*

Files in database submission

Genome browser session
(e.g. UCSC)

## Methodology

Replicates

Sequencing depth

Antibodies

Peak calling parameters

Data quality

Software

# Flow Cytometry

## Plots

Confirm that:

☐ The axis labels state the marker and fluorochrome used (e.g. CD4-FITC).

☐ The axis scales are clearly visible. Include numbers along axes only for bottom left plot of group (a 'group' is an analysis of identical markers).

☐ All plots are contour plots with outliers or pseudocolor plots.

☐ A numerical value for number of cells or percentage (with statistics) is provided.

## Methodology

Sample preparation

Instrument

Software

Cell population abundance

Gating strategy

☐ Tick this box to confirm that a figure exemplifying the gating strategy is provided in the Supplementary Information.

# Magnetic resonance imaging

## Experimental design

Design type

Design specifications

Behavioral performance measures

## Acquisition

Imaging type(s)

Field strength

Sequence & imaging parameters

Area of acquisition

Diffusion MRI ☐ Used ☐ Not used

## Preprocessing

Preprocessing software

Normalization

Normalization template

Noise and artifact removal

Volume censoring

## Statistical modeling & inference

Model type and settings

Effect(s) tested

Specify type of analysis: ☐ Whole brain ☐ ROI-based ☐ Both

Statistic type for inference

(See Eklund et al. 2016)

Correction

## Models & analysis

| n/a | Involved in the study |
|-----|----------------------|
| ☐ ☐ | Functional and/or effective connectivity |
| ☐ ☐ | Graph analysis |
| ☐ ☐ | Multivariate modeling or predictive analysis |

Functional and/or effective connectivity

Graph analysis

Multivariate modeling and predictive analysis

