## [Peer Review File · Nature]

Earliest modern human genomes constrain timing of Neanderthal admixture

Corresponding Author: Ms Arev Sümer

Version 0:

Reviewer comments:

Referee #1

(Remarks to the Author)

The authors have generated a 24x coverage genome from one Upper Palaeolithic individual from Ranis site in Germany (Ranis 13), plus a 20x genome from Zlaty Kun in Czech Republic and five low coverage genomes from other Ranis individuals. They uncover some kinship relationships within this dataset, notably a mother and daughter at Ranis, plus 2h or 3rd degree relatives between two other individuals, and also that Zlaty Kun was a 5-6th degree relative to two Ranis individuals, even if both sites are currently separated by 230 km. The authors show that these individuals diverged earlier than Ust'-Ishim (that is dated to around 44kya) and that they represent so far the earliest human split after the Out of African migration. This evidence allows the authors to estimate the time of Neandertal admixing to a narrow time-window estimated to be between 45 and 49kya.

The generation of novel genome-wide data from early (>40kya) Upper Palaeolithic remains in Eurasia is interesting because of the scarcity of data and the evolutionary consequences that took place in that period, including the Neandertal-modern human admixing and Neandertal's extinction.

I think there are some important points that are a bit overlooked in the current manuscript and maybe the authors could explain/expand/mention/clarify them (even if the paper is not central to some of these questions, the evidences provided can have a consequences beyond the current findings).

1-The date of Neandertal-anatomically modern human (AMH) admixing has been the scope of abundant literature that tried to capture a precise estimate. To mention just a few (many of them also listed in the paper), Sankararaman et al. (2012), Plos Genet, places the date of Neandertal-AMH most likely at 47-65 kya; Sankararaman et al. (2016), Curr. Biol. the Denisovan-AMH at 44-54kya; Vernot and Key (2015) support to pulses of admixing; Moorjani et al. (2016) the Neandertal-AMH at 41-54kya; lasi et al. (2021) at 49kya and Fu et al. (2014) at 52-57 kya. It is a bit unclear what allows now the authors to nail a more precise date; is it the quality of the data, or the estimation methods, or the fact these individuals represent a deeper ancestry split?

2-It has been difficult so far to get an idea on the population size of the earlier Upper Palaeolithic humans into Europe and some works, such as that of Kostenki, made emphasis in the fact individuals analysed were not kinship-related, suggesting large demographic numbers. I think the connection between two sites is interesting because it is suggestive (indirect evidence, of course) of low, or lower than previously assumed, demographics in these expanding groups.

3-An important consequence of a rather late Neandertal-AMH admixing at 45-49kya is the date of the Denisovan-AMH admixing, something that seems to be overlooked in the ms (obviously focused on Europe). Again, there are different estimates of this potential, Asian admixing event, but, since these Upper Palaeolithic individuals and AMH share the same Neandertal pulse, this also implies that Denisovan introgression MUST be posterior to the estimated date (and also AMH need some time to arrive to Asia and Australasia). Is this a correct assumption and if yes, should it be strengthened? There are some dates published for the Denisovan-AMH in East Asia and South Asia (for instance, Yuan et al. 2021, Nat Comm) that could fit in this scenario. However, again, this places the admixing event quite late in human evolutionary history. In this sense, the Tianyuan individual from China, dated to about 40kya does carry a Neandertal signal but apparently no discernible Denisovan signal (although this might be due to the paucity of the data in chr 21).

4-Another consequence that the authors mention is that any Out of Africa migration or migrations prior to these, quite recent dates, left no discernible genetic traces in modern human genomes. This is quite surprising and suggest that not only earlier hominids became extinct, but also AMH that were able to expand Out of Africa before. I accept an explanation of this surprising phenomenon is beyond the scope of this paper, but this is clearly an intriguing finding that will need a future explanation.

5-The authors have used the widely-used 29 years of generation time, but there are some recent estimates such as Wang et

al. (2023), Science Advances, that follow a method based on changes in the mutation spectrum and come out with a figure of 26.9 years. Does this new figure affect the current admixing estimate? Maybe is worth checking it.

Minor points:

Line 109: "human contamination", maybe is better "modern human contamination"

Referee #2

(Remarks to the Author)

*Summary of the key results

This manuscript describes the sequencing and analysis of two high coverage early modern human genomes and five more at lower coverage for from a site in Ranis, Germany and the nearby Zlatý kůň site in Czechia. By comparing these genomes to other ancient humans, Neanderthals, and modern humans, the authors convincingly argue that these individuals represent a closely related group that split early from the Out-of-Africa population, is from the same population as the Zlatý kůň individuals, and is distinct from the Bacho Kiro population that lived at similar times. They then demonstrate that these individuals carry Neanderthal ancestry from a single admixture event (~45-49 kya) that is shared with modern non-Africans.

*Originality and significance: if not novel, please include reference

Genetic data from this critical period in the history of our species are sparse, and thus having two new high coverage genomes provides a significant and powerful perspective from which to answer questions about migration history, genetic relationships, and Neanderthal introgression. The results provide strong and clear support for several previous theories about the relatedness of European early human populations, their relationship to modern non-African individuals. They also help to refine the dating of Neanderthal admixture and support a single admixture event in the history of modern non-African individuals. Overall, these results clarify relationships between early Out-of-Africa modern human populations and provide a valuable model for further study of early human movements out of Africa.

My only critiques about significance are based on data availability and reproducibility. The new genomes are one of the main significant outcomes of this study, and they must be made available both in raw and processed form. The Data Availability section is not clear if the processed genomes will be available. Second, to facilitate reproducibility, I strongly encourage the authors to make their code and data analysis pipelines publicly available on github or a similar platform.

*Data & methodology: validity of approach, quality of data, quality of presentation

The methodologies are sound and clearly presented. The manuscript was easy to understand, and appropriate technical details were provided in the supplementary material.

However, I have several minor suggestions to improve clarity of the presentation of the results:

- The panels in Figure 3C and D are not referenced or explained in the main text. Also, the colors and categories in Figure 3A are not clearly defined in the caption or main text.

- The extended figure references are out of order and some panels are missing references in the text.
- I found several of the extended figures very helpful and informative. I recommend that some be included in main text figures if possible. For example, the F₃ results from Extended Figure 5 and the most supported tree from Extended Figure 7.
- More explanation of the variation in patterns of Neanderthal ancestry correlations across populations in Extended Figure 8A/B would be helpful. Also, why are the patterns across populations different in the correlation (A/B) vs. breakpoint analyses (C/D)?

*Appropriate use of statistics and treatment of uncertainties

In general, I found the statistics to be appropriate and well described. Confidence intervals and/or estimates of uncertainty are generally given. However, the one exception is the Population Continuity section. Providing more technical details on the population continuity analysis in the main text would be helpful. This was the rare section where I found it hard to "fill in the gaps" without doing a deep read of the supplementary material. This is particularly important given the contrasting result with some previous work. In the main text, more is needed on the details on the statistics and methods used for inference and discussion of why these higher coverage data do not reproduce the previous signal of increased sharing with the Buran Kaya III individual.

*Conclusions: robustness, validity, reliability

Clear and appropriate.

*Suggested improvements: experiments, data for possible revision

There is always more that can be done. I had hoped for a bit more analysis on selection after introgression and the potential implications for adaptive trait evolution (SI 20). However, I feel that results support the conclusions, so I do not wish to suggest more work.

*References: appropriate credit to previous work?

Yes.

*Clarity and context: lucidity of abstract/summary, appropriateness of abstract, introduction and conclusions

The manuscript is clear and generally provides sufficient context for interpreting the results. However, I feel would benefit from more context and discussion about how the results presented fit in with other work. For example, it would be helpful to discuss the results on patterns of Neanderthal gene flow in the context of the recent preprint from many of the same authors: Neanderthal ancestry through time: Insights from genomes of ancient and present-day humans

<https://www.biorxiv.org/content/10.1101/2024.05.13.593955v1>

The results are largely consistent, but I am curious if there can be some refinement of the estimates of the duration and timing of introgression.

Referee #3

(Remarks to the Author)

This paper reveals the genetic story of the early modern human populations that colonised Europe around 50 years ago. Its relevance is pivotal for understanding the interbreeding and relationship between late Neanderthals and early modern humans in Eurasia. Recent publications on Ranis and its LRJ culture have proved the presence of modern humans before 40 kya and their temporal and regional overlapping with Neanderthals. This paper is well-structured and pinpoints several key aspects of the human genetic data dated to the Middle to Upper Paleolithic transition that remained unknown so far: 1) identify the kinship and uniparental relationships among the six Ranis individuals where nuclear DNA was preserved; 2) identify the association of two Ranis individuals with Zlatý kůň (Czechia) providing a scenario of limited human population size in Europe at that time which might be conceivable with a single origin; 3) this population named Zlatý kůň/Ranis did not contribute to later hunter-gatherers European populations. 4) homozygosity is being proved due to the small population size rather than consanguinity. And interestingly, 5) it reveals a possible single introgression event after Neanderthals/modern humans admixture dated around 45-49kya with fits with the archaeological evidence. The authors even propose that Zlatý kůň might be an LRJ maker, which would be interested to prove in a near future.

These results offer a European scenario in which the early modern human population was rather small. This Zlatý kůň/Ranis population might represent the earliest split from the out-of-Africa population shortly after a Neanderthal introgression event.

The results presented in this paper carry profound implications for archaeology, paleogenetics, and related disciplines. They provide a deeper understanding of the genetic story of early modern human populations in Europe and their relationship with Neanderthals, enriching our knowledge and opening new avenues for research. This research not only adds to our understanding of the past, but also inspires further exploration and discovery in these fields.

Thus, the conclusions drawn from this research provide valuable insights into human genomes and introgression events in the last 50,000 years. They also state the relationship between material culture and human genomes and pave the way for further exploration and discovery in these fields.

After reviewing the whole manuscript and supplementary materials, I would like to suggest a few improvement points:

-Ref 1 is related to Higham et al. 2014. It refers to the chronology of the Middle to Upper Paleolithic transition in Europe. Although this paper is relevant as it first provided a general European spatiotemporal framework for Neanderthal disappearance, a recent paper in Science Advances by Vidal-Cordasco et al. (2023 - DOI: 10.1126/sci-adv.adi4099) set up more precise chronometric determinations for the spatiotemporal timing of the MUPT in the continent, including new dates and regions not included in the former study. Thus, I recommend including this recent paper.

-It would be helpful for a general reader to define introgression the first time it is mentioned. Now, it is in the second paragraph, around lines 57-58.

-Supplementary Information 2 regarding bone preservation screening explains the use of near-infrared spectroscopy (NIR). NIR screening indicated collagen was sufficiently preserved for radiocarbon dating (~1% minimum requirement). While it is clear the protocol used by Sponheimer et al. (2015), the calibration curve employed to build the models is not stated. It would be helpful to specify it for future studies and researchers.

Ana B. Marín-Arroyo

Version 1:

Reviewer comments:

Referee #1

(Remarks to the Author)

I think the authors have answered the comments outlined in my review and clarified some points regarding the impact of this research on other aspects of the hominin evolution, such as the Denisovans' time for modern human admixture. I agree also that the minor correction on my observation on "modern human contamination" was not needed, as the authors point out.

Referee #2

(Remarks to the Author)

I thank the authors for their responses to my questions and comments.

We would like to thank all three reviewers for their insightful comments that helped us improve our manuscript. Our detailed point-by-point response is shown below, and our responses are marked green. A PDF of the main text and supplementary materials with marked changes is also included in this re-submission.

Referees' comments:

Referee #1 (Remarks to the Author):

The authors have generated a 24x coverage genome from one Upper Palaeolithic individual from Ranis site in Germany (Ranis 13), plus a 20x genome from Zlaty Kun in Czech Republic and five low coverage genomes from other Ranis individuals. They uncover some kinship relationships within this dataset, notably a mother and daughter at Ranis, plus 2nd or 3rd degree relatives between two other individuals, and also that Zlaty Kun was a 5-6th degree relative to two Ranis individuals, even if both sites are currently separated by 230 km. The authors show that these individuals diverged earlier than Ust'-Ishim (that is dated to around 44kya) and that they represent so far the earliest human split after the Out of African migration. This evidence allows the authors to estimate the time of Neandertal admixing to a narrow time-window estimated to be between 45 and 49kya.

The generation of novel genome-wide data from early (>40kya) Upper Palaeolithic remains in Eurasia is interesting because of the scarcity of data and the evolutionary consequences that took place in that period, including the Neandertal-modern human admixing and Neandertal's extinction.

I think there are some important points that are a bit overlooked in the current manuscript and maybe the authors could explain/expand/mention/clarify them (even if the paper is not central to some of these questions, the evidences provided can have a consequences beyond the current findings).

1- The date of Neandertal-anatomically modern human (AMH) admixing has been the scope of abundant literature that tried to capture a precise estimate. To mention just a few (many of them also listed in the paper), Sankararaman et al. (2012), Plos Genet, places the date of Neandertal-AMH most likely at 47-65 kya; Sankararaman et al. (2016), Curr. Biol. the Denisovan-AMH at 44-54kya; Vernot and Key (2015) support to pulses of admixing; Moorjani et al. (2016) the Neandertal-AMH at 41-54kya; Iasi et al. (2021) at 49kya and Fu et al. (2014) at

52-57 kya. It is a bit unclear what allows now the authors to nail a more precise date; is it the quality of the data, or the estimation methods, or the fact these individuals represent a deeper ancestry split?

Segments of Neandertal ancestry decay by each generation. The close proximity of our novel genomes to the actual admixture event as well as the precise radiocarbon dating narrow the confidence intervals for the dating of the Neandertal admixture event substantially. For our admixture time estimate, we incorporated the 95% confidence interval on the radiocarbon date to accurately reflect the uncertainty in dating. The number of generations since admixture was estimated with three different methods and we incorporate the union of the 95% confidence intervals provided by these methods when calculating our admixture date. We added a sentence in line 326 to clarify those points.

2- It has been difficult so far to get an idea on the population size of the earlier Upper Palaeolithic humans into Europe and some works, such as that of Kostenki, made emphasis in the fact individuals analysed were not kinship-related, suggesting large demographic numbers. I think the connection between two sites is interesting because it is suggestive (indirect evidence, of course) of low, or lower than previously assumed, demographics in these expanding groups.

Our estimate based on IBD-sharing between the high-coverage Ranis13 and Zlatý kůň genomes provides us with a direct population size estimate of around 160 individuals within the last 15 generations of when these individuals lived. We have changed the discussion section to further draw attention to this result (line 311):

We inferred a fifth or sixth-degree relationship of this individual to two Ranis individuals and showed that Zlatý kůň falls within the diversity of the Ranis population, in line with a small group size for the Zlatý kůň/Ranis population.

We also refer to the small effective population size as a potential reason for the lack of recent Neandertal ancestry in the Zlatý kůň/Ranis population (line 319):

Further mixing with Neanderthals could have been hindered by a short presence and/or the small size of the Zlatý kůň/Ranis population in Europe.

3- An important consequence of a rather late Neandertal-AMH admixing at 45-49kya is the date of the Denisovan-AMH admixing, something that seems to be overlooked in the ms (obviously focused on Europe). Again, there are different estimates of this potential, Asian admixing event, but, since these Upper Palaeolithic individuals and AMH share the same Neandertal pulse, this also implies that Denisovan introgression MUST be posterior to the estimated date (and also AMH need some time to arrive to Asia and Australasia). Is this a correct assumption and if yes, should it be strengthened? There are some dates published for the Denisovan-AMH in East Asia and South Asia (for instance, Yuan et al. 2021, Nat Comm) that could fit in this scenario.

However, again, this places the admixing event quite late in human evolutionary history. In this sense, the Tianyuan individual from China, dated to about 40kya does carry a Neandertal signal but apparently no discernible Denisovan signal (although this might be due to the paucity of the data in chr 21).

We were unable to detect Denisovan ancestry in the newly sequenced genomes (we now add this result to the Supplementary Information 13 in line 1721), suggesting – as the reviewer points out – that Denisovan ancestry must have entered specific out-of-African populations after the admixture with Neandertals. This insight is also supported by earlier work that found no contribution of Denisovans to ancient western hunter gatherers. We now added a sentence to the discussion (line 334) to draw attention to the fact that the Denisovan admixture post-dates the Neandertal admixture:

Since all populations that carry ancestry from another archaic lineage, the Denisovans, also carry Neanderthal ancestry from this shared event, we can infer that the Denisovan admixture post-dates 45-49 kya.

4- Another consequence that the authors mention is that any Out of Africa migration or migrations prior to these, quite recent dates, left no discernible genetic traces in modern human genomes. This is quite surprising and suggests that not only earlier hominids became extinct, but also AMH that were able to expand Out of Africa before. I accept an explanation of this surprising phenomenon is beyond the scope of this paper, but this is clearly an intriguing finding that will need a future explanation.

We agree that further discussions on the Out of Africa migration are warranted. However, we note that we cannot exclude the possibility that prior Out-of-African migrations are linked to the enigmatic “Basal Eurasian” ancestry detected in present-day Europe and Western Asia or that small quantities of ancestry in later non-African populations originates from such earlier migrations out of Africa (as is debated for Papuan and Australian populations). We now change the last sentence of the main text as following to point this out:

Further study of ancient genomes, fossils and material cultures will be needed to untangle the events surrounding and following Out-of-Africa migrations, such as the origins of the enigmatic Basal Eurasian lineage, and the earliest waves of modern human movements into Europe and Asia.

5- The authors have used the widely-used 29 years of generation time, but there are some recent estimates such as Wang et al. (2023), Science Advances, that follow a method based on changes in the mutation spectrum and come out with a figure of 26.9 years. Does this new figure affect the current admixing estimate? Maybe is worth checking it.

In Wang et al., 2023, the average generation time estimates for the human populations that lived ~43,000 years and ~50,000 years fluctuate between 29.15 and 29.70 years (see supplementary

data table `gentime_estimatesAll.txt`, between `bin_age` 1484.35 and 1712.83), close to the 29 years we use. We note that the assumed generation time does not influence our estimated time of admixture significantly: a one year difference would result in a difference of the estimate of <100 years. We now add a paragraph in the Supplementary Information 13 discussing this on line 1818:

We date the timing of this introgression event using the direct radiocarbon date on Ranis13 and assuming a generation time of 29 years, following previous studies, to 45,024-49,422 years BP. A recent study by Wang et al. 2023 reported generation time estimates of humans in time bins across the past 250,000 years. The average generation time estimate they report for humans who lived between ~43,000 and ~50,000 years fluctuate between 29.15 and 29.70 years (in their supplementary data table named `gentime_estimatesAll.txt`, between `bin_age` 1484.35 and 1712.83). Recalculating the timing of the introgression event with these generation times yields a date from 45,032 to 49,491 years BP, very close to our estimates using a generation time of 29 years.

Minor points:

Line 109: "human contamination", maybe is better "modern human contamination"

The particular method for estimating contamination in line 109 finds deviations from the homozygous state inside regions of homozygosity. It can therefore detect ancient human contamination and also (hypothetically) archaic human contamination (see also Supplementary Section 2 in Posth et al.). We have therefore decided to keep our original phrasing.

Referee #2 (Remarks to the Author):

- Summary of the key results

This manuscript describes the sequencing and analysis of two high coverage early modern human genomes and five more at lower coverage for from a site in Ranis, Germany and the nearby Zlatý kůň site in Czechia. By comparing these genomes to other ancient humans, Neanderthals, and modern humans, the authors convincingly argue that these individuals represent a closely related group that split early from the Out-of-Africa population, is from the same population as the Zlatý kůň individuals, and is distinct from the Bacho Kiro population that lived at similar times. They then demonstrate that these individuals carry Neanderthal ancestry from a single admixture event (~45-49 kya) that is shared with modern non-Africans.

- Originality and significance: if not novel, please include reference

Genetic data from this critical period in the history of our species are sparse, and thus having two new high coverage genomes provides a significant and powerful perspective from which to answer questions about migration history, genetic relationships, and Neanderthal introgression. The results provide strong and clear support for several previous theories about the relatedness of European early human populations, their relationship to modern non-African individuals. They also help to refine the dating of Neanderthal admixture and support a single admixture event in the history of modern non-African individuals. Overall, these results clarify relationships between early Out-of-Africa modern human populations and provide a valuable model for further study of early human movements out of Africa.

My only critiques about significance are based on data availability and reproducibility. The new genomes are one of the main significant outcomes of this study, and they must be made available both in raw and processed form. The Data Availability section is not clear if the processed genomes will be available. Second, to facility reproducibility, I strongly encourage the authors to make their code and data analysis pipelines publicly available on github or a similar platform.

All our raw data have been made publicly available through the ENA Sequence Read Archive (accession: PRJEB78725). In addition, we provide users with processed versions (VCF, recommended filters, and minimally filtered bam files) through our website (<http://ftp.eva.mpg.de/genomes/>). Additional code and data for non-standard analyses have been uploaded to the Max Planck Digital Library or are available through GitHub (D-statistics: <https://doi.org/10.17617/3.0VLEOH> and <https://doi.org/10.17617/3.EGKV28>, Break point analysis: <https://doi.org/10.17617/3.TG4TO4>, PSMC calibration pipeline: <https://github.com/StephanePeyregne/calibratePSMC>, HLA data: <https://doi.org/10.17617/3.GHAALO>). All are cited in the supplementary materials (lines 1285, 1314, 1440, 2175 and 2339).

- Data & methodology: validity of approach, quality of data, quality of presentation

The methodologies are sound and clearly presented. The manuscript was easy to understand, and appropriate technical details were provided in the supplementary material.

However, I have several minor suggestions to improve clarity of the presentation of the results:

- The panels in Figure 3C and D are not referenced or explained in the main text. Also, the colors and categories in Figure 3A are not clearly defined in the caption or main text.

All Figures are now referenced in order in the main text. We defined colors and categories of Figure 3A in the caption.

- The extended figure references are out of order and some panels are missing references in the text.

Extended figures are now sorted in the order of appearance in the main text, and missing references to the panels have been added. For Extended Figure 5, we have only referenced the entire figure as panels should be considered together as tested scenarios.

- I found several of the extended figures very helpful and informative. I recommend that some be included in main text figures if possible. For example, the F_3 results from Extended Figure 5 and the most supported tree from Extended Figure 7.

We included the subset of the $f\beta$ -matrix with the individuals that lived before 40,000 years BP as Panel B in the main Figure 2. However, we prefer leaving the most supported qpGraph model in the extended figures so that it is shown together with the other tested models. We present the main findings from the qpGraph and other analyses in the summary tree in Figure 4.

- More explanation of the variation in patterns of Neanderthal ancestry correlations across populations in Extended Figure 8A/B would be helpful. Also, why are the patterns across populations different in the correlation (A/B) vs. breakpoint analyses (C/D)?

There are at least three major factors that contribute to variation between populations and differences between the correlation and breakpoint analyses. First, the correlation analysis is based on the SGDP data whereas the breakpoint analysis uses a combined dataset incorporating 1000 Genomes and HGDP data. This means that the number and identity of the individuals in the input dataset, and also the composition of super-populations, do not match. Second, the methods to detect archaic segments differs between the two approaches: breakpoint analysis aims at detecting switches in Neanderthal ancestry with high accuracy based fixed derived Neanderthal variants whereas the correlation analysis uses posterior probabilities of the archaic segment detection method *admixfrog*. Lastly, the presence of Denisovan ancestry in some populations will inevitably lead to some misclassification of introgressed segments, although the impact is likely more severe for the breakpoint than the correlation analysis since Neanderthal ancestry informative sites used in the former analysis will partly detect Denisovan ancestry. However, some misclassification is unavoidable as the available Denisovan genome is more diverged from the introgressed Denisovan segments than available Neanderthal genomes are from introgressed Neanderthal segments. Future analysis may be able to quantify the impact of these factors to determine whether differences between populations are due to population structure. For this study, we focussed on identifying a signal of shared Neanderthal ancestry that we can conclusively show (e.g. Supplementary Figure 16.6 and 16.8).

- Appropriate use of statistics and treatment of uncertainties

In general, I found the statistics to be appropriate and well described. Confidence intervals and/or estimates of uncertainty are generally given. However, the one exception is the Population Continuity section. Providing more technical details on the population continuity analysis in the

main text would be helpful. This was the rare section where I found it hard to “fill in the gaps” without doing a deep read of the supplementary material. This is particularly important given the contrasting result with some previous work. In the main text, more is needed on the details on the statistics and methods used for inference and discussion of why these higher coverage data do not reproduce the previous signal of increased sharing with the Buran Kaya III individual.

We now provide additional information for the population continuity analyses using the high coverage genomes, including the statistics with the Buran Kaya III individuals in the main text lines 175 and 188, along with data on the Max Planck Digital Library (available through <https://doi.org/10.17617/3.0VLEOH>).

Furthermore, we repeated the analyses using a different approach, which is more similar to the one followed by Bennett et al. We added the results at the end of Supplementary Information 9, from line 1289 on. In short, we again could not replicate the results from Bennett et al., 2023 and did not observe signals suggesting continuity from the Zlatý kůň/Ranis population to Buran Kaya III individuals. The data and code used for this analysis is also available through <https://doi.org/10.17617/3.EGKV28>.

We note that the results of Bennett et al. are also inconsistent with the earlier work from Posth et al. Most notably, some of the f4-statistics involving Fournol 85 yield significant ($|Z|>3$) results in Bennett et al. (2023) while identical comparisons in Posth et al. (2023) were not significant. This difference is surprising given that Bennett et al. filtered the data more and retained fewer sites in these comparisons (listed here with the labels given in the supplementary tables of both publications):

Publication	Comparison				F4	Z-score	nBABA	nABBA	nSNPs
Posth et al.	Mbuti.DG	ZlatyKun	FRL	Russia_Kostenki14	-0.000386	-0.491	18425	18566	365167
Bennett et al.	Mbuti	CZ_IUP_ZlatýKůň	FR_Gra_Fournol85	RU_UP_Kostenki14	-0.002132	-2.433	2403	2569	78211
Posth et al.	Mbuti.DG	ZlatyKun	FRL	Russia_Sungghir.SG	-0.001124	-1.809	19252	19686	386101
Bennett et al.	Mbuti	CZ_IUP_ZlatýKůň	FR_Gra_Fournol85	RU_UP_Sungghir3	-0.002752	-3.538	2647	2909	95202
Posth et al.	Mbuti.DG	ZlatyKun	Czech_Vestonice	FRL	0.000301	0.49	18578	18465	375638
Bennett et al.	Mbuti	CZ_IUP_ZlatýKůň	FR_Gra_Fournol85	Vestonice	-0.003258	-3.864	2066	2283	66499

- Conclusions: robustness, validity, reliability: Clear and appropriate.

- Suggested improvements: experiments, data for possible revision

There is always more that can be done. I had hoped for a bit more analysis on selection after introgression and the potential implications for adaptive trait evolution (SI 20). However, I feel that results support the conclusions, so I do not wish to suggest more work.

We agree that selection is a topic that could be further explored using these new data. However, we feel that we cannot give this topic the attention in the paper that it would require. We are confident that the data will be used in studies of adaptive evolution by us and others in the future.

- References: appropriate credit to previous work? Yes.

- Clarity and context: lucidity of abstract/summary, appropriateness of abstract, introduction and conclusions

The manuscript is clear and generally provides sufficient context for interpreting the results. However, I feel would benefit from more context and discussion about how the results presented fit in with other work. For example, it would be helpful to discuss the results on patterns of Neanderthal gene flow in the context of the recent preprint from many of the same authors:

Neandertal ancestry through time: Insights from genomes of ancient and present-day humans
<https://www.biorxiv.org/content/10.1101/2024.05.13.593955v1>

The results are largely consistent, but I am curious if there can be some refinement of the estimates of the duration and timing of introgression.

Our admixture time estimate is exclusively based on the radiocarbon date of Ranis13 together with various estimates of the number of generations that passed for this individual since the admixture. To clarify this, we now add a sentence to discussion (line 327, see the response to the first point of the first reviewer). In contrast, the paper by Iasi et al. estimates the time point of admixture from a large number of radiocarbon dated ancient individuals that are younger than those studied in this work. It is indeed reassuring that this study comes to similar conclusions. Future extensions of the work of Iasi et al. could incorporate the data from Ranis individuals to further refine the estimate.

Referee #3 (Remarks to the Author):

This paper reveals the genetic story of the early modern human populations that colonised Europe around 50 years ago. Its relevance is pivotal for understanding the interbreeding and relationship between late Neanderthals and early modern humans in Eurasia. Recent publications

on Ranis and its LRJ culture have proved the presence of modern humans before 40 kya and their temporal and regional overlapping with Neanderthals. This paper is well-structured and pinpoints several key aspects of the human genetic data dated to the Middle to Upper Paleolithic transition that remained unknown so far: 1) identify the kinship and uniparental relationships among the six Ranis individuals where nuclear DNA was preserved; 2) identify the association of two Ranis individuals with Zlatý kůň (Czechia) providing a scenario of limited human population size in Europe at that time which might be conceivable with a single origin; 3) this population named Zlatý kůň/Ranis did not contribute to later hunter-gatherers European populations. 4) homozygosity is being proved due to the small population size rather than consanguinity. And interestingly, 5) it reveals a possible single introgression event after Neanderthals/modern humans admixture dated around 45-49kya with fits with the archaeological evidence. The authors even propose that Zlatý kůň might be an LRJ maker, which would be interested to prove in a near future.

These results offer a European scenario in which the early modern human population was rather small. This Zlatý kůň/Ranis population might represent the earliest split from the out-of-Africa population shortly after a Neanderthal introgression event.

The results presented in this paper carry profound implications for archaeology, paleogenetics, and related disciplines. They provide a deeper understanding of the genetic story of early modern human populations in Europe and their relationship with Neanderthals, enriching our knowledge and opening new avenues for research. This research not only adds to our understanding of the past, but also inspires further exploration and discovery in these fields.

Thus, the conclusions drawn from this research provide valuable insights into human genomes and introgression events in the last 50,000 years. They also state the relationship between material culture and human genomes and pave the way for further exploration and discovery in these fields.

After reviewing the whole manuscript and supplementary materials, I would like to suggest a few improvement points:

- Ref 1 is related to Higham et al. 2014. It refers to the chronology of the Middle to Upper Paleolithic transition in Europe. Although this paper is relevant as it first provided a general European spatiotemporal framework for Neanderthal disappearance, a recent paper in Science Advances by Vidal-Cordasco et al. (2023 - DOI: 10.1126/sciadv.adi4099) set up more precise chronometric determinations for the spatiotemporal timing of the MUPT in the continent, including new dates and regions not included in the former study. Thus, I recommend including this recent paper.

We thank the reviewer for this suggestion and have added the citation.

- It would be helpful for a general reader to define introgression the first time it is mentioned. Now, it is in the second paragraph, around lines 57-58.

We have now added a short definition of introgression at first occurrence in the main text.

- Supplementary Information 2 regarding bone preservation screening explains the use of near-infrared spectroscopy (NIR). NIR screening indicated collagen was sufficiently preserved for radiocarbon dating (~1% minimum requirement). While it is clear the protocol used by Sponheimer et al. (2015), the calibration curve employed to build the models is not stated. It would be helpful to specify it for future studies and researchers.

Our method is based on the protocol and dataset reported in Sponheimer et al 2019 but supplemented by an in-house dataset of NIR scans of bones with known collagen yields which is updated as new samples are pretreated. We have updated the Supplementary Information 2 text to reflect this (line 257-258).

Ana B. Marín-Arroyo

Additional changes

- We added a sentence in the main text line 254 to refer to Figure 3C.
- In the main text methods section, line 621, we changed “introgressed fragments” to “introgressed segments” to keep our nomenclature consistent.
- Additional sequencing platform information was added in the Supplementary Information 3, line 368 and Supplementary Information 4, line 456.
- We recalculated the Neandertal ancestry proportions on X-chromosomes using a different ascertainment (Extended Archaic Ascertainment). A publication on the ArchaicX ascertainment that we originally used is currently in preparation, but will require more time until published. We instead ascertained sites by requiring four high-coverage archaic human genomes (Vindija33.19, Chagyrskaya8, Denisova5, Denisova3) to differ from all Sub-Saharan Africans (ESN, GWD, LWK, MLS and YRI from 1000 Genomes Project and an Mbuti genome from HGDP (HGDP0456)) (Iasi et al. 2024, bioRxiv). We note that the length of the two segments detected on the X-chromosome of Ranis13 remained unchanged. The mean percentage of Neandertal ancestry on the X chromosome of Ust'-Ishim and Ranis13 remains much lower than that of the autosomal chromosomes with the new ascertainment. We also removed the reference to the Bacho Kiro estimates as these estimates use a different ascertainment and may not be comparable. Changes affect Supplementary Information 13,

lines 1700 and the paragraph starting with line 1731 and the Methods section of the main text line 608.

- We modified two sentences on lines 1269-1271 to clarify the message and to make the previous sentence grammatically correct.
- We made species and genome names italic, while de-italicizing the names of the softwares/tools.
- We add Jörg Orschiedt as one of the co-authors on Supplementary Information 1.
- We modified the order of the lineages in main text Figure 4 to reflect the geography and updated the figure caption.
- We added text (page 11, lines 279-288 & 306) to Supplementary Information 2 (Collagen extraction and dating) to include elemental and stable isotope values for the two collagen extracts dated in the study, which are well-known quality indicators for the suitability of collagen extracts for ¹⁴C dating.
- During revision we also found and removed small typographical errors in the supplementaries and main text.